# DFRD: Data-Free Robustness Distillation for Heterogeneous Federated Learning

**Kangyang Luo**[1], **Shuai Wang**[1], **Yexuan Fu**[1], **Xiang Li**[1,*], **Yunshi Lan**[1], **Ming Gao**[1,2]

School of Data Science & Engineering[1]
KLATASDS-MOE in School of Statistics[2]
East China Normal University
Shanghai, China
{52205901003, 51215903058,51215903042}@stu.ecnu.edu.cn
{xiangli, yslan, mgao}@dase.ecnu.edu.cn

## Abstract

Federated Learning (FL) is a privacy-constrained decentralized machine learning paradigm in which clients enable collaborative training without compromising private data. However, how to learn a robust global model in the data-heterogeneous and model-heterogeneous FL scenarios is challenging. To address it, we resort to data-free knowledge distillation to propose a new FL method (namely DFRD). DFRD equips a conditional generator on the server to approximate the training space of the local models uploaded by clients, and systematically investigates its training in terms of *fidelity*, *transferability* and *diversity*. To overcome the catastrophic forgetting of the global model caused by the distribution shifts of the generator across communication rounds, we maintain an exponential moving average copy of the generator on the server. Additionally, we propose dynamic weighting and label sampling to accurately extract knowledge from local models. Finally, our extensive experiments on various image classification tasks illustrate that DFRD achieves significant performance gains compared to SOTA baselines. Our code is here: https://anonymous.4open.science/r/DFRD-0C83/.

## 1 Introduction

With the surge of data, deep learning algorithms have made significant progress in both established and emerging fields [1–3]. However, in many real-world applications (e.g., mobile devices [4], IoT [5], and autonomous driving [6], etc.), data is generally dispersed across different clients (i.e., data silos). Owing to the high cost of data collection and strict privacy protection regulations, the centralized training that integrates data together is prohibited [7, 8]. Driven by this reality, Federated Learning (FL) [9, 10] has gained considerable attention in industry and academia as a promising distributed learning paradigm that allows multiple clients to participate in the collaborative training of a global model without access to their private data, thereby ensuring the basic privacy.

Despite its remarkable success, the inevitable hurdle that plagues FL research is the vast heterogeneity among real-world clients [11]. Specifically, the distribution of data among clients may be non-IID (identical and independently distributed), resulting in **data heterogeneity** [42, 12, 13, 8, 41]. It has been confirmed that the vanilla FL method FedAvg [14] suffers from *client drift* in this case, which leads to severe performance degradation. To ameliorate this issue, a plethora of modifications [15–21, 34, 39] for FedAvg focus on regularizing the objectives of the local models to align the global optimization objective. All of the above methods follow the widely accepted assumption of model homogeneity, where local models have to share the same architecture as the global model. Nevertheless,

---

*corresponding author

37th Conference on Neural Information Processing Systems (NeurIPS 2023).

when deploying FL systems, different clients may have distinct hardware and computing resources, and can only train the model architecture matching their own resource budgets [22, 24], resulting in **model heterogeneity**. In this case, to enable the FL systems with model homogeneity, on the one hand, clients with low resource budgets, which may be critical for enhancing the FL systems, will be discarded at the expense of training bias [43, 8, 23]. On the other hand, keeping a low complexity for the global model to accommodate all clients leads to performance drop due to the limited model capacity [30]. Therefore, the primary challenge of model-heterogeneous FL is how to conduct model aggregation of heterogeneous architectures among clients to enhance the inclusiveness of federated systems. To solve this challenge, existing efforts fall broadly into two categories: knowledge distillation (KD)-based methods [40, 45, 44, 25–27] and partial training (PT)-based methods [30, 28, 29, 31], yet each of them has its own limitations. Concretely, KD-based methods require additional public data to align the logits outputs between the global model (student) and local models (teachers). But the desired public data is not always available in practice and the performance may decrease dramatically if the apparent disparity in distributions exists between public data and clients' private data [33]. PT-based methods send width-based sub-models to clients, which are extracted by the server from a larger global model according to each client's resource budget, and then aggregate these trained sub-models to update the global model. PT-based methods can be considered as an extension of FedAvg to model-heterogeneous scenarios, which means they are implementation-friendly and computationally efficient, but they also suffer from the same adverse effects from data heterogeneity as FedAvg or even more severely. In a word, how to learn a robust global model in FL with both data and model heterogeneity is a highly meaningful and urgent problem.

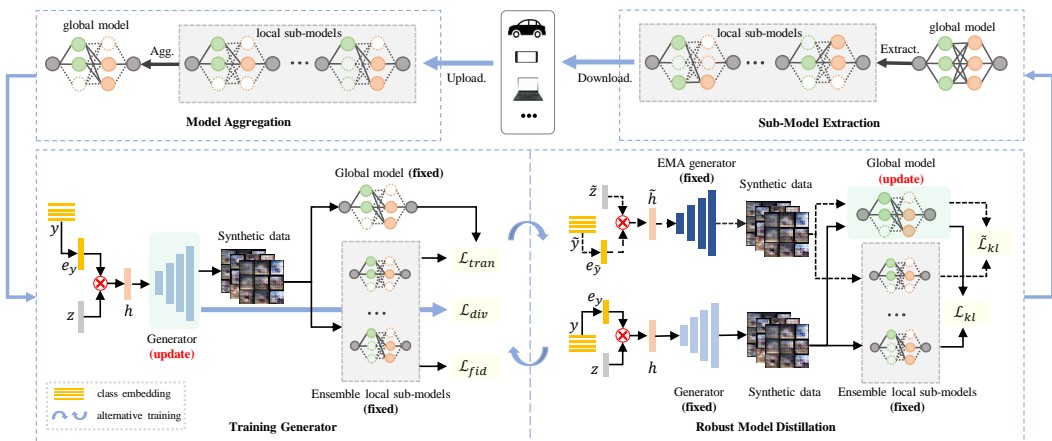

Figure 1: The full pipeline for DFRD combined with a PT-based method. DFRD works on the server and contains two phases, *training generator* and *robust model distillation*, where $\mathcal{L}_{tran}, \mathcal{L}_{div}, \mathcal{L}_{fid}$ and $\mathcal{L}_{kl}, \widetilde{\mathcal{L}}_{kl}$ are the loss objectives of the conditional generator and the global model, respectively.

To this end, we systematically investigate the training of a robust global model in FL with both data and model heterogeneity with the aid of data-free knowledge distillation (DFKD) [35, 36, 53, 59]. See the related work in Appendix A for more DFKD methods. Note that the strategy of integrating DFKD to FL is not unique to us. Recently, FedFTG [48] leverages DFKD to fine-tune the global model in model-homogeneous FL to overcome data heterogeneity, and DENSE [49] aggregates knowledge from heterogeneous local models based on DFKD to train a global model for one-shot FL. They all equip the server, which possesses powerful hardware and computing resources, with a generator to approximate the training space of the local models (teachers), and train the generator and the global model (student) in an adversarial manner. However, the local models uploaded per communication round are not only architecturally heterogeneous but also trained on non-IID distributed private data in the situation of both data and model heterogeneity. *In this case, the generator tends to deviate from the real data distribution. Also, its output distribution may undergo large shifts (i.e., distribution shifts) across communication rounds, causing the global model to catastrophically forget useful knowledge learned in previous rounds and suffer from performance degradation.* To confront the mentioned issues, we propose a novel **D**ata-**F**ree **R**obust **D**istillation FL method called DFRD, which utilizes a conditional generator to generate synthetic data and thoroughly studies how to effectively and accurately simulate the local models' training space in terms of *fidelity*, *transferability* and

*diversity* [48, 49]. To mitigate catastrophic forgetting of the global model, an exponential moving average (EMA) copy of the conditional generator is maintained on the server to store previous knowledge learned from the local models. The EMA generator, along with the current generator, provides training data for updates of the global model. Also, we propose dynamic weighting and label sampling to aggregate the logits outputs of the local models and sample labels respectively, thereby properly exploring the knowledge of the local models. We revisit FedFTG and DENSE, and argue that DFRD as a fine-tuning method (similar to FedFTG) can significantly enhance the global model. So, we readily associate the PT-based methods in model-heterogeneous FL with the ability to rapidly provide a preliminary global model, which will be fine-tuned by DFRD. We illustrate the schematic for DFRD as a fine-tuning method based on a PT-based method in Fig. 1. Although FedFTG and DENSE can also be applied to fine-tune the global model from the PT-based methods after simple extensions, we empirically find that they do not perform as well, and the performance of the global model is even inferior to that of local models tailored to clients' resource budgets.

Our main contributions of this work are summarized as follows. First, we propose a new FL method termed DFRD that enables a robust global model in both data and model heterogeneity settings with the help of DFKD. Second, we systematically study the training of the conditional generator w.r.t. *fidelity*, *transferability* and *diversity* to ensure the generation of high-quality synthetic data. Additionally, we maintain an EMA generator on the server to overcome the global model's catastrophic forgetting caused by the distribution shifts of the generator. Third, we propose dynamic weighting and label sampling to accurately extract the knowledge of local models. At last, our extensive experiments on six real-world image classification datasets verify the superiority of DFRD.

## 2 Notations and Preliminaries

**Notations.** In this paper, we focus on the centralized setup that consists of a central server and $N$ clients owning private labeled datasets $\{(\boldsymbol{X}_i, \boldsymbol{Y}_i)\}_{i=1}^{N}$, where $\boldsymbol{X}_i = \{\boldsymbol{x}_i^b\}_{b=1}^{n_i}$ follows the data distribution $\mathcal{D}_i$ over feature space $\mathcal{X}_i$, i.e., $\boldsymbol{x}_i^b \sim \mathcal{D}_i$, and $\boldsymbol{Y}_i = \{y_i^b\}_{b=1}^{n_i} \subset [C] := \{1, \cdots, C\}$ denotes the ground-truth labels of $\boldsymbol{X}_i$. And $C$ refers to the total number of labels. Remarkably, the heterogeneity for FL in our focus includes both data heterogeneity and model heterogeneity. **For the former**, we consider the same feature space, yet the data distribution may be different among clients, that is, label distribution skewness in clients (i.e., $\mathcal{X}_i = \mathcal{X}_j$ and $\mathcal{D}_i \neq \mathcal{D}_j, \forall i \neq j, i, j \in [N]$). **For the latter**, each client $i$ holds an on-demand local model $f_i$ parameterized by $\boldsymbol{\theta}_i$. Due to the difference in resource budgets, the model capacity of each client may vary, i.e., $|\boldsymbol{\theta}_i| \neq |\boldsymbol{\theta}_j|, \exists i \neq j, i, j \in [N]$. In PT-based methods, we define a confined width capability $R_i \in (0, 1]$ according to the resource budget of client $i$, which is the proportion of nodes extracted from each layer in the global model $f$. Note that $f$ is parameterized by $\boldsymbol{\theta}$, and $|\boldsymbol{a}|$ denotes the number of elements in vector $\boldsymbol{a}$.

**PT-based method** is a solution for model-heterogeneous FL, which strives to extract a matching width-based slimmed-down sub-model from the global model as a local model according to each client's budget. As with FedAvg, it requires the server to periodically communicate with the clients. In each round, there are two phases: local training and server aggregation. In local training, each client trains the sub-model received from the server utilizing the local optimizer. In server aggregation, the server collects the heterogeneous sub-models and aggregates them by straightforward selective averaging to update the global model, as follows [28–31]:

$$\boldsymbol{\theta}_{[l,k]}^t = \frac{1}{\sum_{j \in \mathcal{S}_t} p_j} \sum_{i \in \mathcal{S}_t} p_i \boldsymbol{\theta}_{i,[l,k]}^t, \tag{1}$$

where $\mathcal{S}_t$ is a subset sampled from $[N]$ and $p_i$ is the weight of client $i$, which generally indicates the size of data held by client $i$. At round $t$, $\boldsymbol{\theta}_{[l,k]}^t$ denotes the $k^{th}$ parameter of layer $l$ of the global model and $\boldsymbol{\theta}_{i,[l,k]}^t$ denotes the parameter $\boldsymbol{\theta}_{[l,k]}^t$ updated by client $i$. We can clearly see that Eq. (1) independently calculates the average of each parameter for the global model according to how many clients update that parameter in round $t$. Instead, the parameter remains unchanged if no clients update it. Notably, if $|\boldsymbol{\theta}_i^t| = |\boldsymbol{\theta}^t|$ for any $i \in [N]$, PT-based method becomes FedAvg. The key to PT-based method is to select $\boldsymbol{\theta}_i^t$ from the global model $\boldsymbol{\theta}^t$ when given $R_i$. And existing sub-model extraction schemes fall into three categories: *static* [28, 29], *random* [30] and *rolling* [31].

# 3 Proposed Method

In this section, we detail the proposed method DFRD. We mainly work on considering DFRD as a fine-tuning method to enhance the PT-based methods, thus enabling a robust global model in FL with both data and model heterogeneity. Fig. 1 visualizes the training procedure of DFRD combined with a PT-based method, consisting of four stages on the server side: *training generator*, *robust model distillation*, *sub-model extraction* and *model aggregation*. Note that *sub-model extraction* and *model aggregation* are consistent with that in the PT-based methods, so we detail the other two stages. Moreover, we present pseudocode for DFRD in Appendix C.

## 3.1 Training Generator

At this stage, we aim to train a well-behaved generator to capture the training space of local models uploaded from active clients. Specifically, we consider a conditional generator $G(\cdot)$ parameterized by $\boldsymbol{w}$. It takes as input a random noise $\boldsymbol{z} \in \mathbb{R}^d$ sampled from standard normal distribution $\mathcal{N}(\boldsymbol{0}, \boldsymbol{I})$, and a random label $y \in [C]$ sampled from label distribution $p(y)$, i.e., the probability of sampling $y$, thus generating the synthetic data $\boldsymbol{s} = G(\boldsymbol{h} = o(\boldsymbol{z}, y), \boldsymbol{w})$. Note that $o(\boldsymbol{z}, y)$ represents the merge operator of $\boldsymbol{z}$ and $y$. To the best of our knowledge, synthetic data generated by a well-trained generator should satisfy several key characteristics: *fidelity*, *transferability*, and *diversity* [48, 49]. Therefore, in this section, we construct the loss objective from the referred aspects to ensure the quality and utility of $G(\cdot)$.

**Fidelity.** To commence, we study the fidelity of the synthetic data. Specifically, we expect $G(\cdot)$ to simulate the training space of the local models to generate the synthetic dataset with a similar distribution to the original dataset. To put it differently, we want the synthetic data $\boldsymbol{s}$ to approximate the training data with label $y$ without access to clients' training data. To achieve it, we form the fidelity loss $\mathcal{L}_{fid}$ at logits level:

$$\mathcal{L}_{fid} = CE(\sum_{i \in \mathcal{S}_t} \tau_{i,y} f_i(\boldsymbol{s}, \boldsymbol{\theta}_i), y), \tag{2}$$

where $CE$ denotes the cross-entropy function, $f_i(\boldsymbol{s}, \boldsymbol{\theta}_i)$ is the logits output of the local model from client $i$ when $\boldsymbol{s}$ is given, $\tau_{i,y}$ dominates the weight of logits from different clients $\{i|i \in \mathcal{S}_t\}$ when $y$ is given. And $\mathcal{L}_{fid}$ is the cross-entropy loss between the weighted average logits $\sum_{i \in \mathcal{S}_t} \tau_{i,y} f_i(\boldsymbol{s}, \boldsymbol{\theta}_i)$ and the label $y$. By minimizing $\mathcal{L}_{fid}$, $\boldsymbol{s}$ is enforced to be classified into label $y$ with a high probability, thus facilitating the fidelity of $\boldsymbol{s}$.

In reality, the conditional generator $G(\cdot)$ easily generates synthetic data with low classification errors (i.e. $\mathcal{L}_{fid}$ close to 0) as the training proceeds. This may cause the synthetic data to fall into a space far from the decision boundary of the ensemble model (i.e., $\sum_{i \in \mathcal{S}_t} \tau_{i,y} f_i(\cdot, \boldsymbol{\theta}_i)$) if only $\mathcal{L}_{fid}$ is optimized, as shown in the synthetic data represented by red circles in Fig. 2 (a). Note that $d_S$ and $d_T$ denote the decision boundaries of the global model (student) and the ensemble model (teacher), respectively. An obvious observation is that the red circles are correctly classified on the same side of the two decision boundaries (i.e., $d_S$ and $d_T$), making it difficult to transfer teacher's knowledge to student. We next explore how to augment the transferability of the synthetic data to ameliorate this pitfall.

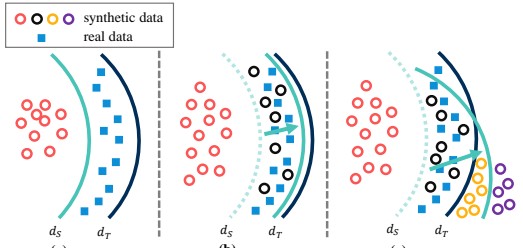

Figure 2: The visualization of synthetic data and decision boundaries of global model (student) and ensemble model (teacher). *Left panel*: synthetic data (red circles) are far away from the decision boundary $d_T$. *Middle panel*: synthetic data (black circles) near the decision boundaries $d_T$. *Right panel*: synthetic data (yellow and purple circles) cross over the decision boundary $d_T$.

**Transferability** is intended to guide $G(\cdot)$ in generating synthetic data that moves the decision boundary of the global model towards that of the ensemble model, such as synthetic data with black circles in Fig. 2 (b). However, during the training of $d_S$ to approach $d_T$, we find that $G(\cdot)$ can easily generate two other types of synthetic data, the yellow and purple circles in Fig. 2 (c). Both of them are misclassified by the ensemble model ($d_T$), while the yellow circles are correctly classified and

the purple circles are misclassified by the global model ($d_S$). For the conditional generator $G(\cdot)$ that takes label information as one of the inputs, yellow and purple circles can mislead the generator, thereby leading to $d_S$ approximating $d_T$ with a large deviation, as shown in Fig. 2 (c). Based on the above observation, we reckon that the synthetic data $\boldsymbol{s} = G(\boldsymbol{h} = o(\boldsymbol{z}, y), \boldsymbol{w})$ is useful if it is classified as $y$ by the ensemble model but classified not as $y$ by the global model. To realize it, we maximize the logits discrepancy between the global model and the ensemble model on synthetic data with black circles by leveraging Kullback-Leibler divergence loss, which takes the form:

$$\mathcal{L}_{tran} = -\varepsilon \cdot KL(\sum_{i \in \mathcal{S}_t} \tau_{i,y} f_i(\boldsymbol{s}, \boldsymbol{\theta}_i), f(\boldsymbol{s}, \boldsymbol{\theta})), \tag{3}$$

where $KL$ is Kullback-Leibler divergence function and $f(\boldsymbol{s}, \boldsymbol{\theta})$ denotes the logits output of the global model on $\boldsymbol{s}$ with label $y$. Note that $\varepsilon = 1$ if $\arg\max f(\boldsymbol{s}, \boldsymbol{\theta}) \neq y$ and $\arg\max \sum_{i \in \mathcal{S}_t} \tau_{i,y} f_i(\boldsymbol{s}, \boldsymbol{\theta}_i) = y$ hold, otherwise $\varepsilon = 0$. ($\Diamond$)

We would like to point out that the existing works [48] and [49] are in line with our research perspective on the transferability of generator, which aims to generate more synthetic data with black circles. However, they do not carefully frame learning objective for enhancing the transferability of generator. Concretely, [48] does not consider the type of synthetic data, i.e., $\varepsilon = 1$ always holds, thus inducing the generation of synthetic data with yellow and purple circles. ($\triangle$) [49] focuses on synthetic data satisfying $\arg\max f(\boldsymbol{s}, \boldsymbol{\theta}) \neq \arg\max \sum_{i \in \mathcal{S}_t} \tau_{i,y} f_i(\boldsymbol{s}, \boldsymbol{\theta}_i)$, but enables the generation of synthetic data with purple circles yet. ($\triangledown$) [2]

**Diversity.** Although we enable $G(\cdot)$ to generate synthetic data that falls around the real data by optimizing $\mathcal{L}_{fid}$ and $\mathcal{L}_{tran}$, the diversity of synthetic data is insufficient. Due to the fact that the generator may get stuck in *local equilibria* as the training proceeds, model collapse occurs [57, 58]. In this case, the generator may produce similar data points for each class with little diversity. Also, the synthetic data points may not differ significantly among classes. This causes the empirical distribution estimated by $G(\cdot)$ to cover only a small manifold in the real data space, and thus only partial knowledge of the ensemble model is extracted. To alleviate this issue, we introduce a diversity loss $\mathcal{L}_{div}$ with label information to increase the diversity of synthetic data as follows:

$$\mathcal{L}_{div} = \exp\left(-\sum_{j,k \in [B]} \|\boldsymbol{s}_j - \boldsymbol{s}_k\|_2 * \|\boldsymbol{h}_j - \boldsymbol{h}_k\|_2 / B^2\right), \tag{4}$$

where $B$ denotes the batch size and $\boldsymbol{s}_{j/k} = G(\boldsymbol{h}_{j/k} = o(\boldsymbol{z}_{j/k}, y_{j/k}), \boldsymbol{w})$. Intuitively, $\mathcal{L}_{div}$ takes $\|\boldsymbol{h}_j - \boldsymbol{h}_k\|_2$ as a weight, and then multiplies it by the corresponding $\|\boldsymbol{s}_j - \boldsymbol{s}_k\|_2$ in each batch $B$, thus imposing a larger weight on the synthetic data points pair ($\boldsymbol{s}_j$ and $\boldsymbol{s}_k$) at the more distant input pair ($\boldsymbol{h}_j$ and $\boldsymbol{h}_k$). Notably, we merge the random noise $\boldsymbol{z}$ with label $y$ as the input of $G(\cdot)$ to overcome spurious solutions [53]. Further, we propose a multiplicative merge operator, i.e., $o(\boldsymbol{z}, y) = \boldsymbol{z} \times \mathcal{E}(y)$, where $\mathcal{E}$ is a trainable embedding and $\times$ means vector element-wise product. We find that our merge operator enables synthetic data with more diversity compared to others, possibly because the label information is effectively absorbed into the stochasticity of $\boldsymbol{z}$ by multiplying them when updating $\mathcal{E}$. See Section 4.3 for more details and empirical justification.

Combining $\mathcal{L}_{fid}$, $\mathcal{L}_{tran}$ and $\mathcal{L}_{div}$, the overall objective of the generator can be formalized as follows:

$$\mathcal{L}_{gen} = \mathcal{L}_{fid} + \beta_{tran} \cdot \mathcal{L}_{tran} + \beta_{div} \cdot \mathcal{L}_{div}, \tag{5}$$

where $\beta_{tran}$ and $\beta_{div}$ are tunable hyper-parameters. Of note, the synthetic data generated by a well-trained generator should be visually distinct from the real data for privacy protection, while it can capture the common knowledge of the local models to ensure similarity to the real data distribution for utility. More privacy protection is discussed in Appendices I and J.

### 3.2 Robust Model Distillation

Now we update the global model. Normally, the global model attempts to learn as much as possible logits outputs of the ensemble model on the synthetic data generated by the generator based on knowledge distillation [40, 46, 48, 49]. The updated global model and the ensemble model are then served to train $G(\cdot)$ with the goal of generating synthetic data that maximizes the mismatch

---

[2]Note that $\triangle$, $\triangledown$ and $\Diamond$ denote the strategies for the transferability of generator in [48], in [49] and in this paper, respectively.

between them in terms of logits outputs (see *transferability* discussed in the previous section). This adversarial game enables the generator to rapidly explore the training space of the local models to help knowledge transfer from them to the global model. However, it also leads to dramatic shifts in the output distribution of $G(\cdot)$ across communication rounds under heterogeneous FL scenario (i.e., distribution shifts), causing the global model to catastrophically forget useful knowledge gained in previous rounds. To tackle the deficiency, we propose to equip the server with a generator $\widetilde{G}(\cdot)$ parameterized by $\widetilde{w}$ that is an exponential moving average (EMA) copy of $G(\cdot)$. Its parameters at the $t^{th}$ communication round are computed by

$$\widetilde{w}^t = \lambda \cdot \widetilde{w}^{t-1} + (1 - \lambda) \cdot w^t, \tag{6}$$

where $\lambda \in (0, 1)$ is the momentum. We can easily see that the parameters of $\widetilde{G}(\cdot)$ vary very little compared to those of $G(\cdot)$ over communication rounds, if $\lambda$ is close to 1. We further utilize synthetic data from $\widetilde{G}(\cdot)$ as additional training data for the global model outside of $G(\cdot)$, mitigating the huge exploratory distribution shift induced by the large update of $G(\cdot)$ and achieving stable updates of the global model. Particularly, we compute the Kullback-Leibler divergence between logits of the ensemble model and the global model on the synthetic data points $s = G(h = o(z, y), w)$ and $\widetilde{s} = \widetilde{G}(\widetilde{h} = o(\widetilde{z}, \widetilde{y}), \widetilde{w})$ respectively, which is formulated as follows:

$$\mathcal{L}_{md} = \mathcal{L}_{kl} + \alpha \widetilde{\mathcal{L}}_{kl} = KL(f(s, \theta), \sum_{i \in \mathcal{S}_t} \tau_{i,y} f_i(s, \theta_i)) + \alpha \cdot KL(f(\widetilde{s}, \theta), \sum_{i \in \mathcal{S}_t} \tau_{i,\widetilde{y}} f_i(\widetilde{s}, \theta_i)), \tag{7}$$

where $\alpha$ is a tunable hyper-parameter for balancing different loss items.

**Dynamic Weighting and Label Sampling.** So far, how to determine $\tau_{i,y}$ and $p(y)$ is unclear. The appropriate $\tau_{i,y}$ and $p(y)$ are essential for effective extraction of knowledge from local models. For clarity, we propose dynamic weighting and label sampling, i.e., $\tau_{i,y} = n_{i,t}^y / n_{\mathcal{S}_t,t}^y$ and $p(y) = n_{\mathcal{S}_t,t}^y / \sum_{y \in [C]} n_{\mathcal{S}_t,t}^y$, where $n_{\mathcal{S}_t,t}^y = \sum_{j \in [\mathcal{S}_t]} n_{j,t}^y$ and $n_{i,t}^y$ denotes the number of data with label $y$ involved in training on client $i$ at round $t$. Due to space limitations, see Appendix F for their detail study and experimental justification.

## 4 Experiments

### 4.1 Experimental Settings

**Datasets.** In this paper, we evaluate different methods with six real-world image classification task-related datasets, namely Fashion-MNIST [69] (FMNIST in short), SVHN [70], CIFAR-10, CIFAR-100 [71], Tiny-imageNet[3] and Food101 [73]. We detail the six datasets in Appendix B. To simulate data heterogeneity across clients, as in previous works [34, 37, 38], we use Dirichlet process $Dir(\omega)$ to partition the training set for each dataset, thereby allocating local training data for each client. It is worth noting that $\omega$ is the concentration parameter and smaller $\omega$ corresponds to stronger data heterogeneity.

**Baselines.** We compare DFRD to FedFTG [48] and DENSE [49], which are the most relevant methods to our work. To verify the superiority of DFRD, on the one hand, DFRD, FedFTG and DENSE are directly adopted on the server to transfer the knowledge of the local models to a randomly initialized global model. We call them collectively **data-free methods**. On the other hand, they are utilized as **fine-tuning methods** to improve the global model's performance after computing weighted average per communication round. In this case, in each communication round, the preliminary global model is obtained using FedAvg [14] in FL with homogeneous models, whereas in FL with heterogeneous models, the PT-based methods, including HeteroFL [29], Federated Dropout [30] (FedDp for short) and FedRolex [31], are employed to get the preliminary global model. [4]

**Configurations.** Unless otherwise stated, all experiments are performed on a centralized network with $N = 10$ active clients. We set $\omega \in \{0.01, 0.1, 1.0\}$ to mimic different data heterogeneity scenarios. To simulate model-heterogeneous scenarios, we formulate exponentially distributed budgets for a given $N$: $R_i = [\frac{1}{2}]^{\min\{\sigma, \lfloor \frac{\rho \cdot i}{N} \rfloor\}} (i \in [N])$, where $\sigma$ and $\rho$ are both positive integers. We fix $\sigma = 4$

---

[3]http://cs231n.stanford.edu/tiny-imagenet-200.zip

[4]As an example, DFRD+FedRelox indicates that DFRD is used as a fine-tuning method to improve the performance of FedRelox, and others are similar. See Tables 1 and 2 for details.

and consider $\rho \in \{5, 10, 40\}$. See Appendix D for more details. Unless otherwise specified, we set $\beta_{tran}$ and $\beta_{div}$ both to 1 in *training generator*, while in *robust model distillation*, we set $\lambda = 0.5$ and $\alpha = 0.5$. And all baselines leverage the same setting as ours. Due to space limitations, see Appendix E for the full experimental setup.

**Evaluation Metrics.** We evaluate the performance of different FL methods by local and global test accuracy. To be specific, for local test accuracy (*L.acc* for short), we randomly and evenly distribute the test set to each client and harness the test set on each client to verify the performance of local models. In terms of global test accuracy (*G.acc* for short), we employ the global model on the server to evaluate the global performance of different FL methods via utilizing the original test set. Note that *L.acc* is reported in **round brackets**. To ensure reliability, we report the average for each experiment over 3 different random seeds.

Table 1: Top test accuracy (%) of distinct methods across $\omega \in \{0.01, 0.1, 1.0\}$ on different datasets.

| Alg.s | FMNIST | | | SVHN | | | CIFAR-10 | | | CIFAR-100 | | |
|---|---|---|---|---|---|---|---|---|---|---|---|---|
| | $\omega=1.0$ | $\omega=0.1$ | $\omega=0.01$ | $\omega=1.0$ | $\omega=0.1$ | $\omega=0.01$ | $\omega=1.0$ | $\omega=0.1$ | $\omega=0.01$ | $\omega=1.0$ | $\omega=0.1$ | $\omega=0.01$ |
| DENSE | 13.99±4.39 (74.26±2.64) | 13.96±1.33 (30.27±1.10) | 13.83±2.88 (14.82±3.55) | 16.12±2.08 (52.67±1.68) | 14.53±3.56 (25.95±0.42) | 13.06±5.40 (13.09±1.64) | 10.47±1.21 (46.89±0.97) | 10.28±0.60 (22.67±0.66) | 10.96±1.93 (13.03±1.56) | 1.22±0.10 (21.68±0.23) | 1.11±0.03 (13.61±0.30) | 1.11±0.02 (8.73±0.09) |
| FedFTG | 48.10±1.61 (73.96±2.51) | 44.44±1.30 (30.33±1.43) | 26.96±8.33 (14.79±3.61) | 18.58±1.41 (54.75±0.63) | 17.42±2.03 (26.39±0.38) | 15.44±4.12 (13.15±1.63) | 28.73±1.03 (47.67±0.92) | 21.97±1.18 (22.95±0.64) | 15.18±0.72 (13.06±1.54) | 1.05±0.09 (22.67±0.44) | 1.06±0.10 (14.01±0.49) | 1.11±0.09 (8.82±0.21) |
| DFRD | **65.38±0.72** (74.44±2.47) | **52.55±1.61** (30.44±1.42) | **36.63±6.52** (14.87±3.64) | **24.13±0.96** (55.48±1.56) | **18.53±0.42** (26.87±0.11) | **18.18±0.56** (12.83±1.90) | **33.82±1.81** (48.33±1.17) | **24.58±0.59** (23.21±0.70) | **20.59±2.93** (14.18±1.53) | **3.21±0.36** (23.31±0.39) | **1.55±0.13** (14.15±0.32) | **1.36±0.06** (8.93±0.13) |
| FedAvg | 89.71±0.10 (84.57±0.63) | 83.24±1.41 (62.48±5.86) | 60.89±5.24 (29.81±13.98) | 70.63±1.95 (54.21±2.54) | 58.12±1.26 (24.66±0.42) | 32.22±1.16 (12.57±1.81) | 67.16±0.98 (49.82±1.47) | 56.36±2.09 (21.92±0.83) | 32.92±7.40 (12.37±1.71) | 57.31±0.09 (47.01±0.99) | 49.50±0.51 (25.33±0.88) | 39.22±1.08 (11.25±0.52) |
| +DENSE | 90.13±0.14 (84.81±0.60) | 84.10±1.55 (62.80±5.76) | 63.36±6.24 (29.37±13.83) | 74.48±1.48 (58.02±2.65) | 62.41±2.36 (26.23±0.33) | 32.84±12.37 (12.72±1.82) | 69.73±0.60 (53.46±1.58) | 57.48±3.13 (22.54±0.76) | 35.85±7.22 (12.41±1.72) | 58.43±0.05 (48.00±0.78) | 50.84±0.60 (26.19±0.90) | 40.24±1.04 (11.87±0.58) |
| +FedFTG | 91.15±0.23 (85.88±0.70) | 84.93±1.40 (63.14±4.64) | 64.80±6.56 (**29.89±14.08**) | 73.91±1.78 (56.99±2.65) | 61.12±1.69 (25.82±0.28) | 37.07±6.55 (12.69±1.80) | 68.85±0.96 (51.05±1.59) | 58.17±2.02 (22.39±0.66) | 33.93±7.69 (12.39±1.70) | 58.81±1.39 (48.06±1.79) | 50.70±1.11 (26.40±0.47) | 40.50±1.17 (12.15±0.61) |
| +DFRD | **91.65±0.09** (**86.16±0.53**) | **85.66±1.21** (**63.79±4.29**) | **65.23±7.17** (29.58±13.45) | **77.84±1.39** (**60.31±2.51**) | **65.79±2.27** (**27.26±0.20**) | **39.25±11.71** (**13.96±1.48**) | **71.54±0.66** (**53.59±1.19**) | **61.34±1.21** (**23.00±0.68**) | **40.77±7.47** (**13.49±1.57**) | **61.49±0.28** (**49.97±0.58**) | **53.95±0.67** (**29.10±0.87**) | **43.47±1.05** (**14.09±1.09**) |

## 4.2 Results Comparison

**Impacts of varying $\omega$.** We study the performance of different methods at different levels of data heterogeneity on FMNIST, SVHN, CIFAR-10 and CIFAR-100, as shown in Table 1. One can see that the performance of all methods degrades severely as $\omega$ decreases, with DFRD being the only method that is robust while consistently leading other baselines with an overwhelming margin w.r.t. *G.acc*. Also, Fig. 3 (a)-(b) show that the learning efficiency of DFRD consistently beats other baselines (see Fig. 8-9 in Appendix H for complete curves). Notably, DFRD, FedFTG and DENSE as fine-tuning methods uniformly surpass FedAvg w.r.t. *G.acc* and *L.acc*. However, their global test accuracies suffer from dramatic deterioration or even substantially worse than that of FedAvg when they act as data-free methods. We conjecture that FedAvg aggregates the knowledge of local models more effectively than data-free methods. Also, when DFRD is used to fine-tune FedAvg, it can significantly enhance the global model, yet improve the performance of local models to a less extent.

**Impacts of different $\rho$.** We explore the impacts of different model heterogeneity distributions on different methods with SVHN, CIFAR-10, Tiny-ImageNet and FOOD101. A higher $\rho$ means more clients with $\frac{1}{16}$-width capacity w.r.t. the global model. From Table 2, we can clearly see that the performance of all methods improves uniformly with decreasing $\rho$, where DFRD consistently and overwhelmingly dominates other baselines in terms of *G.acc*. Specifically, DFRD improves *G.acc* by an average of 11.07% and 7.54% on SVHN and CIFAR-10 respectively, compared to PT-based methods (including HeteroFL, FedDp and FedRolex). Meanwhile, DFRD uniformly and significantly outstrips FedFTG and DENSE w.r.t. *G.acc*. The selected learning curve shown in Fig. 3 (c) also verifies the above statement (see Fig. 10-12 in Appendix H for more results). The above empirical results show that DFRD not only is robust to varying $\rho$, but also has significantly intensified effects on the global model for different PT-based methods. However, the results on Tiny-ImageNet and FOOD101 indicate that PT-based methods suffer from inferior test accuracy. Although DFRD improves their test accuracy, the improvement is slight. Notably, DFRD improves negligibly over PT-based methods when all clients exemplify $\frac{1}{16}$-width capability. We thus argue that weak clients performing complex image classification tasks learn little useful local knowledge, resulting in the inability to provide effective information for the global model.

## 4.3 Ablation Study

In this section, we carefully demonstrate the efficacy and indispensability of core modules and key parameters in our method on SVHN, CIFAR-10 and CIFAR-100. Thereafter, we resort to FedRolex+DFRD to yield all results. For SVHN (CIFAR-10, CIFAR-100), we set $\omega = 0.1$ (0.1, 1.0)

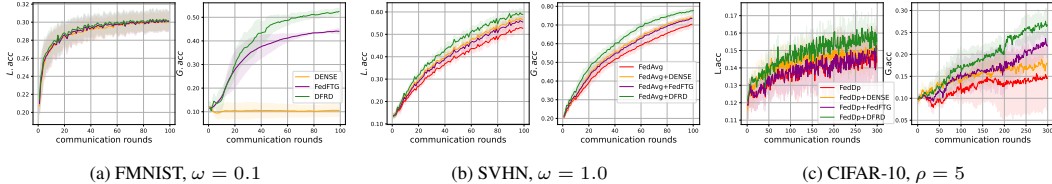

(a) FMNIST, $\omega = 0.1$      (b) SVHN, $\omega = 1.0$      (c) CIFAR-10, $\rho = 5$

Figure 3: Accuracy curves selected of DFRD and baselines on FMNIST, SVHN and CIFAR-10.

Table 2: Top test accuracy (%) of distinct methods across $\rho \in \{5, 10, 40\}$ on different datasets.

| Alg.s | SVHN $\rho=5$ | SVHN $\rho=10$ | SVHN $\rho=40$ | CIFAR-10 $\rho=5$ | CIFAR-10 $\rho=10$ | CIFAR-10 $\rho=40$ | Tiny-ImageNet $\rho=5$ | Tiny-ImageNet $\rho=10$ | Tiny-ImageNet $\rho=40$ | FOOD101 $\rho=5$ | FOOD101 $\rho=10$ | FOOD101 $\rho=40$ |
|---|---|---|---|---|---|---|---|---|---|---|---|---|
| HeteroFL | 29.56±17.60 (27.63±5.35) | 23.07±12.92 (30.90±5.70) | 17.37±9.50 (25.28±1.13) | 21.00±3.28 (23.94±2.63) | 13.89±3.73 (**24.70±0.80**) | 11.16±1.95 (26.55±1.29) | 8.27±0.21 (6.72±0.13) | 1.03±0.20 (5.46±0.33) | 0.73±0.11 (6.90±0.23) | 9.61±0.97 (7.79±0.49) | 1.52±0.17 (9.12±0.66) | 1.16±0.07 (12.05±0.37) |
| +DENSE | 31.16±15.65 (28.62±4.62) | 27.48±9.61 (31.51±4.27) | 20.47±6.24 (24.88±1.82) | 21.63±2.83 (24.05±2.52) | 16.64±2.86 (24.34±0.73) | 12.79±0.28 (**26.80±1.17**) | 8.44±0.32 (6.74±0.28) | 1.07±0.16 (5.86±0.80) | 0.75±0.08 (7.15±0.35) | 9.78±0.85 (7.98±0.43) | 1.90±0.39 (8.99±0.66) | 1.24±0.11 (11.99±0.33) |
| +FedFTG | 32.22±15.66 (29.10±5.00) | 26.92±8.86 (**31.56±4.45**) | 19.02±4.51 (25.33±1.30) | 22.29±4.01 (24.34±2.64) | 18.79±4.58 (24.47±0.85) | 15.69±2.91 (26.46±1.12) | 8.38±0.20 (6.63±0.32) | 1.08±0.17 (6.01±0.69) | 0.74±0.10 (7.11±0.51) | 9.85±1.07 (7.88±0.47) | 1.94±0.44 (8.74±0.56) | 1.26±0.17 (11.87±0.16) |
| +DFRD | **42.77±12.60** (**29.41±5.53**) | **30.17±7.26** (**31.56±4.56**) | **24.82±6.70** (**26.14±0.47**) | **24.30±1.59** (24.70±1.58) | **23.78±1.77** (24.35±0.67) | **19.10±0.78** (26.75±1.34) | **9.27±0.14** (**8.09±0.20**) | **1.50±0.12** (**8.13±0.33**) | **0.83±0.01** (**10.29±0.18**) | **11.54±0.32** (**9.28±0.07**) | **3.05±0.73** (**10.94±0.07**) | **1.58±0.23** (**14.39±0.02**) |
| FedDP | 28.99±17.35 (16.96±5.50) | 24.17±18.36 (14.95±2.59) | 20.03±13.90 (15.48±0.87) | 17.36±3.13 (15.47±1.06) | 12.52±1.41 (15.48±0.87) | 11.08±1.83 (15.83±1.07) | 6.33±0.78 (2.84±0.30) | 1.15±0.17 (1.14±0.28) | 0.97±0.13 (0.97±0.10) | 7.69±0.58 (3.42±0.22) | 1.84±0.59 (1.70±0.32) | 1.75±0.28 (1.70±0.28) |
| +DENSE | 31.15±22.51 (18.39±6.36) | 26.38±18.60 (15.73±4.00) | 22.33±14.46 (14.65±2.93) | 19.66±1.15 (15.63±0.95) | 14.55±1.50 (15.51±0.73) | 12.22±1.89 (15.61±1.24) | 6.54±0.39 (2.75±0.29) | 1.23±0.25 (1.17±0.33) | 1.03±0.10 (0.99±0.11) | 8.19±0.91 (3.59±0.23) | 1.98±0.33 (1.69±0.44) | 1.66±0.49 (1.69±0.25) |
| +FedFTG | 32.30±20.90 (18.00±6.10) | 28.55±14.65 (15.04±2.80) | 26.63±12.64 (14.39±1.60) | 23.67±2.70 (15.68±1.24) | 13.80±0.41 (15.51±0.73) | 12.17±1.87 (15.61±1.24) | 6.32±0.53 (2.84±0.25) | 1.22±0.19 (1.16±0.31) | 0.98±0.21 (0.96±0.17) | 7.93±0.49 (3.44±0.22) | 2.08±0.24 (1.65±0.45) | 1.69±0.25 (1.68±0.28) |
| +DFRD | **41.48±8.11** (**19.02±4.75**) | **37.30±12.22** (**17.19±0.28**) | **32.89±11.65** (**16.14±0.30**) | **27.93±3.63** (**16.69±0.82**) | **20.24±1.44** (**16.35±0.95**) | **19.42±4.37** (**15.97±0.95**) | **9.01±0.42** (**3.83±0.07**) | **1.59±0.21** (**1.56±0.20**) | **1.05±0.23** (**1.25±0.08**) | **11.28±0.44** (**4.62±0.13**) | **2.71±0.24** (**2.14±0.07**) | **1.94±0.12** (**1.82±0.01**) |
| FedRolex | 34.71±13.68 (14.20±2.94) | 23.48±13.09 (13.85±2.60) | 22.39±15.28 (13.59±2.11) | 21.11±1.76 (16.99±0.67) | 16.57±4.40 (16.12±0.90) | 14.37±2.91 (17.11±1.29) | 9.29±0.32 (5.33±0.21) | 5.55±0.40 (2.73±0.09) | 2.50±0.33 (1.81±0.29) | 10.27±1.33 (6.86±0.12) | 5.14±3.41 (3.37±1.14) | 3.22±0.52 (2.95±0.17) |
| +DENSE | 36.58±12.53 (17.51±2.52) | 26.69±13.11 (14.49±1.96) | 24.34±14.81 (14.04±1.95) | 23.72±5.48 (17.16±0.67) | 19.65±1.47 (15.81±0.62) | 16.44±1.89 (17.26±1.34) | 9.33±0.06 (5.16±0.18) | 5.40±0.40 (2.76±0.01) | 2.40±0.20 (1.82±0.33) | 10.83±0.78 (6.95±0.26) | 7.54±0.46 (3.86±0.44) | 3.07±0.53 (2.99±0.11) |
| +FedFTG | 38.07±12.27 (17.64±2.54) | 25.53±13.84 (14.51±2.21) | 24.06±14.86 (14.03±1.91) | 22.66±5.24 (17.16±1.03) | 17.79±2.67 (16.25±1.56) | 14.77±2.03 (17.70±1.14) | 9.36±0.23 (5.18±0.16) | 5.68±0.34 (2.75±0.05) | 2.43±0.12 (1.81±0.23) | 10.66±0.79 (6.85±0.11) | 8.13±0.82 (3.75±0.48) | 3.06±0.48 (2.92±0.19) |
| +DFRD | **46.30±10.12** (**18.44±1.34**) | **34.78±9.19** (**15.99±1.53**) | **32.86±15.54** (**15.17±0.66**) | **26.68±1.21** (**17.51±0.33**) | **25.57±1.37** (**16.74±0.72**) | **19.86±2.76** (**17.77±1.16**) | **10.93±0.05** (**6.11±0.02**) | **6.80±0.11** (**3.35±0.04**) | **2.68±0.19** (**2.22±0.08**) | **12.70±0.79** (**8.02±0.08**) | **10.58±0.29** (**4.86±0.16**) | **3.59±0.05** (**3.34±0.15**) |

and $\rho = 10$ $(10, 5)$. Note that we also study the performance of DFRD with different numbers of clients and stragglers, see Appendix G for details.

**Impacts of different transferability constraints.** It can be observed from Table 3 that our proposed transferability constraint $\Diamond$ uniformly beats $\triangle$ and $\triangledown$ w.r.t. global test performance over three datasets. This means that $\Diamond$ can guide the generator to generate more effective synthetic data, thereby improving the test performance of the global model. Also, we conjecture that generating more synthetic data like those with the black circles (see Fig. 2) may positively impact on the performance of local models, since $\Diamond$ also slightly and consistently trumps other competitors w.r.t. $L.acc$.

**Impacts of varying merge operators.** To look into the utility of the merge operator, we consider multiple merge operators, including $mul$, $add$, $cat$, $ncat$ and $none$. Among them, $mul$ is $o(\boldsymbol{z}, y) = \boldsymbol{z} \times \mathcal{E}(y)$, $add$ is $o(\boldsymbol{z}, y) = \boldsymbol{z} + \mathcal{E}(y)$, $cat$ is $o(\boldsymbol{z}, y) = [\boldsymbol{z}, \mathcal{E}(y)]$, $ncat$ is $o(\boldsymbol{z}, y) = [\boldsymbol{z}, y]$ and $none$ is $o(\boldsymbol{z}, y) = \boldsymbol{z}$. From Table 4, we can observe that our proposed $mul$ significantly outperforms other competitors in terms of $G.acc$. This suggests that $mul$ can better exploit the diversity constraint to make the generator generate more diverse synthetic data, thus effectively fine-tuning the global model. Also, the visualization of the synthetic data in Appendix I validates this statement.

Table 3: Test accuracy (%) comparison among different transferability constraints over SVHN and CIFAR-10/100.

| T. C. | SVHN | CIFAR-10 | CIFAR-100 |
|---|---|---|---|
| $\triangle$ | 34.09±11.45 (15.38±2.50) | 23.45±1.13 (15.91±1.24) | 26.62±2.39 (12.76±0.51) |
| $\triangledown$ | 33.89±11.22 (15.28±2.59) | 24.27±1.82 (16.45±1.00) | 27.18±1.78 (12.86±0.38) |
| $\Diamond$ | **34.78±9.19** (**15.99±1.53**) | **25.57±1.37** (**16.74±0.72**) | **28.08±0.94** (**13.03±0.16**) |

Table 4: Test accuracy (%) comparison among different merger operators over SVHN and CIFAR-10/100.

| M. O. | SVHN | CIFAR-10 | CIFAR-100 |
|---|---|---|---|
| $mul$ | **34.78±9.19** (15.99±1.53) | **25.57±1.37** (16.74±0.72) | **28.08±0.94** (**13.03±0.16**) |
| $add$ | 32.83±8.86 (**16.04±1.25**) | 23.36±1.57 (16.55±0.82) | 23.46±2.04 (12.22±0.44) |
| $cat$ | 32.84±10.80 (15.97±1.46) | 21.78±0.20 (16.58±0.72) | 24.14±2.31 (12.27±0.52) |
| $ncat$ | 29.58±9.29 (13.37±2.87) | 20.93±0.80 (**16.87±0.62**) | 23.38±1.43 (12.30±0.16) |
| $none$ | 27.79±10.70 (13.99±2.82) | 19.64±1.08 (16.53±0.83) | 20.95±2.32 (12.81±2.32) |

**Necessity of each component for DFRD.** We report the test performance of DFRD after divesting some modules and losses in Table 5. Here, EMA indicates the exponential moving average copy of the generator on the server. We can evidently observe that removing the EMA generator leads to a significant drop in $G.acc$, which implies that it can generate effective synthetic data for the global model. The reason why the EMA generator works is that it avoids catastrophic forgetting of the global model and ensures the stability of the global model trained in heterogeneous FL. We display synthetic data in Appendix I that further corroborates the above claims.

Meanwhile, we perform the leave-one-out test to explore the contributions of $\mathcal{L}_{tran}$ and $\mathcal{L}_{div}$ to DFRD separately, and further report the test results of removing them simultaneously. From Table 5, deleting either $\mathcal{L}_{tran}$ or $\mathcal{L}_{div}$ adversely affects the performance of DFRD. In addition, their joint absence further exacerbates the degradation of $G.acc$. This suggests that $\mathcal{L}_{tran}$ and $\mathcal{L}_{div}$ are vital for the training of the generator. Interestingly, $\mathcal{L}_{div}$ benefits more to the global model than $\mathcal{L}_{tran}$. We speculate that the diversity of synthetic data is more desired by the global model under the premise of ensuring the fidelity of synthetic data by optimizing $\mathcal{L}_{fid}$.

Table 5: Impact of each component in DFRD.

|  | SVHN | CIFAR-10 | CIFAR-100 |
|---|---|---|---|
| baseline | **34.78**±**9.19** (**15.99**±**1.53**) | **25.57**±**1.37** (**16.74**±**0.72**) | **28.08**±**0.94** (**13.03**±**0.16**) |
| -EMA | 26.97±12.28 (14.17±3.09) | 19.80±2.25 (16.55±0.57) | 24.57±0.93 (12.23±0.07) |
| -$\mathcal{L}_{tran}$ | 29.30±9.25 (14.24±3.08) | 22.97±1.91 (16.33±1.16) | 27.28±0.46 (12.79±0.46) |
| -$\mathcal{L}_{div}$ | 27.68±9.75 (14.26±3.00) | 22.12±1.03 (16.46±1.15) | 26.94±1.60 (12.81±0.41) |
| -$\mathcal{L}_{tran}$, -$\mathcal{L}_{div}$ | 20.32±1.93 (13.65±1.13) | 21.97±2.48 (16.52±1.39) | 25.50±0.91 (12.30±0.10) |

**Varying $\beta_{tran}$ and $\beta_{div}$.** We explore the impacts of $\beta_{tran}$ and $\beta_{div}$ on SVHN and CIFAR-10. We select $\beta_{tran}$ and $\beta_{div}$ from $\{0.25, 0.50, 0.75, 1.00, 1.25, 1.50\}$. From Fig. 4, we can see that DFRD maintains stable test performance among all selections of $\beta_{tran}$ and $\beta_{div}$ over SVHN. At the same time, $G.acc$ fluctuates slightly with the

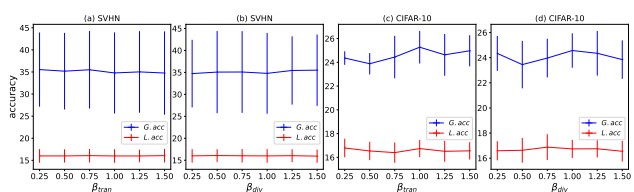

Figure 4: Test accuracy (%) with varying $\beta_{tran}$ and $\beta_{div}$.

increases of $\beta_{tran}$ and $\beta_{div}$ on CIFAR-10. Besides, we observe that the worst $G.acc$ in Fig. 4 outperforms the baseline with the best $G.acc$ in Table 2. The above results indicate that DFRD is not sensitive to choices of $\beta_{tran}$ and $\beta_{div}$ over a wide range.

**Varying $\alpha$ and $\lambda$.** In order to delve into the effect of the EMA generator on DFRD in more details, we perform grid testing on the choices of control parameters $\alpha$ and $\lambda$ over SVHN and CIFAR-10. We set $\alpha \in \{0.25, 0.50, 0.75, 1.00, 1.25, 1.50\}$ and $\lambda \in \{0.1, 0.3, 0.5, 0.7, 0.9\}$. It can be observed from Fig. 5 that high global test accuracies on

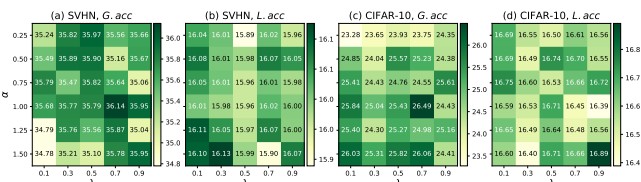

Figure 5: Test accuracy (%) with varying ($\alpha$, $\lambda$).

SVHN are mainly located in the region of $\alpha < 1.25$ and $\lambda > 0.5$, while on CIFAR-10 they are mainly located in the region of $\alpha > 0.25$ and $\lambda < 0.9$. According to the above results, we deem that the appropriate $\alpha$ and $\lambda$ in a specific task is essential for the utility of the EMA generator. Notably, high local test accuracies mainly sit in regions that are complementary to those of high global test accuracies, suggesting that pursuing high $G.acc$ and $L.acc$ simultaneously seems to be a dilemma. How to ensure high $G.acc$ and $L.acc$ simultaneously in the field of FL is an attractive topic that is taken as our future work.

## 5 Conclusion

In this paper, we propose a new FL method called DFRD, which aims to learn a robust global model in the data-heterogeneous and model-heterogeneous scenarios with the aid of DFKD. To ensure the utility, DFRD considers a conditional generator and thoroughly studies its training in terms of *fidelity*,

*transferability* and *diversity*. Additionally, DFRD maintains an EMA generator to augment the global model. Furthermore, we propose dynamic weighting and label sampling to accurately extract the knowledge of local models. At last, we conduct extensive experiments to verify the superiority of DFRD. Due to space constraints, we discuss in detail the **limitations** and **broader impacts** of our work in Appendixes J and K, respectively.

## 6 Acknowledgments

This work has been supported by the National Natural Science Foundation of China under Grant No.U1911203, and the National Natural Science Foundation of China under Grant No.62377012.

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
