# Appendix

## A Related Work

**Heterogeneous Federated Learning.** Federated Learning (FL) has emerged as a de facto machine learning area and received rapidly increasing research interests from the community. FedAvg, the classic distributed learning framework for FL, is first proposed by McMahan et al. [14]. Although FedAvg provides a practical and simple solution for aggregation, it still suffers from performance degradation or even fails to run due to the vast heterogeneity among real-world clients [13, 26].

On the one hand, the distribution of data among clients may be non-IID (identical and independently distributed), resulting in **data heterogeneity** [12, 42, 8, 41, 13]. To ameliorate this issue, a myriad of modifications for FedAvg have been proposed [15–21, 32, 34, 39]. For example, FedProx [15] constrains local updates by adding a proximal term to the local objectives. Scaffold [16] uses control variate to augment the local updates. FedDyn [17] dynamically regularizes the objectives of clients to align global and local objectives. Moon [39] corrects the local training by conducting contrastive learning in model-level. GradMA [34] takes inspiration from continual learning to simultaneously rectify the server-side and client-side update directions. FedMLB [18] architecturally regularizes the local objectives via online knowledge distillation.

On the other hand, due to different hardware and computing resources [22, 24], clients require to customize model capacities, which poses a more practical challenge: **model heterogeneity**. Preceding methods, which are developed under the assumption that the local models and the global model have to share the same architecture, cannot work on heterogeneous models. To address this problem, existing efforts fall broadly into two categories: knowledge distillation (KD)-based methods [44, 45, 40, 25–27] and partial training (PT)-based methods [30, 28, 29, 31]. For **the former**, FedDF [40] conducts KD based on logits averages to refine the network of the same size or several prototypes. FedGKT [45] aggregates knowledge from clients and transfers it to a larger global model on the server based on KD. FedMD [44] performs communication between clients by using KD, which requires the server to collect logits of the public data on each local model and compute the average as the consensus for updates. Fed-ET [26] aims to train a large global model with smaller local models using a weighted consensus distillation scheme with diversity regularization. RHFL [27] aligns the logits outputs of the local models by KD based on public data. However, most of the mentioned methods are data-dependent, that is, they often depend on access to public data which may not always be available in practice. For **the latter**, PT-based methods adaptively extract a matching width-based slimmed-down sub-model from the global model as a local model according to each client's budget, thus averting the requirements for public data. As with FedAvg, PT-based methods require the server to periodically communicate with the clients. Therefore, PT-based methods can be considered as an extension of FedAvg to model-heterogeneous scenarios. Existing PT-based methods focus on how to extract width-based sub-models from the global model. For example, HeteroFL [29] and FjORD [28] propose a static extraction scheme in which sub-models are always extracted from a specified part of the global model. Federated Dropout [30] proposes a random extraction scheme, which randomly extracts sub-models from the global model. FedRolex [31] proposes a rolling sub-model extraction scheme, where the sub-model is extracted from the global model using a rolling window that advances in each communication round. Going beyond the aforementioned methods, some works overcome model heterogeneity from other perspectives. For example, InclusiveFL [63] and DepthFL [64] consider depth-based sub-models with respect to the global model. Also, FedProto [33] and FedHeNN [23] work on personalized FL with model heterogeneity.

**Data-Free Knowledge Distillation (DFKD).** DFKD methods are promising, which transfer knowledge from the teacher model to another student model without any real data. The generation of synthetic data that facilitates knowledge transfer is crucial in DFKD methods. Existing DFKD methods can be broadly classified into non-adversarial and adversarial training methods. Non-adversarial training methods [35, 59–62] exploit certain heuristics to search for synthetic data similar to the original training data. For example, DAFL [35] and ZSKD [61] regard the predicted probabilities and confidences of the teacher model as heuristics. Adversarial training methods [51–53] take advantage of adversarial learning to explore the distribution space of the original data. They take the quality and/or diversity of the synthetic data as important objectives. For example, CMI [51] improves the diversity of synthetic data by leveraging inverse contrastive loss. DeepInversion [52] augments

synthetic data by regularizing the distribution of Batch Norm to ensure visual interpretability. Recently, some efforts [53–56] have delved into the problem of catastrophic forgetting in adversarial settings. Among them, MAD [53] mitigates the generator's forgetting of previous knowledge by maintaining an exponential moving average copy of the generator. Compared to [54–56], MAD is more memory efficient, adapts better to the student update, and is more stable for a continuous stream of synthetic samples. As such, in this paper we opt to maintain an exponential moving average copy of the generator to achieve robust model distillation.

**Federated Learning with DFKD.** The fact that DFKD does not need real data coincides with the requirement of FL to protect clients' private data. A series of recent works [47–50] mitigate heterogeneity in FL by using DFKD. For example, FedGEN [47] uploads the last layer of the local models to the server and trains a conditional generator using DFKD to boost the local model update of each client. FedFTG [48] leverages DFKD to train a conditional generator on the server based on local models, and then fine-tune the global model by KD in model-homogeneous FL. DENSE [49] aggregates knowledge from heterogeneous local models based on DFKD to train a global model for one-shot FL. Most Recently, FEDCVAE-KD [50] reframes the local learning task using a conditional variational autoencoder, and leverages KD to compress the ensemble of client decoders into a global decoder. Then, it trains a global model using the synthetic data generated by the global decoder. Note that FEDCVAE-KD considers the one-shot FL.

## B    Datasets

In Table 6, we provide information about the image size, the number of classes (#class), and the number of training/test samples (#train/#test) of the datasets used in our experiment. Of note, we have used Resize() in PyTorch to scale up and down the image size of FMNIST and FOOD101, respectively.

Table 6: Statistics of the datasets used in our experiments.

| Dataset | Image size | #class | #train | #test |
|---------|-----------|--------|--------|-------|
| FMNIST | 1*32*32 | 10 | 60000 | 10000 |
| SVHN | | 10 | 50000 | 10000 |
| CIFAR-10 | 3*32*32 | 10 | 50000 | 10000 |
| CIFAR-100 | | 100 | 73257 | 26032 |
| Tiny-ImageNet | | 200 | 100000 | 10000 |
| FOOD101 | 3*64*64 | 101 | 75750 | 25250 |

## C    Pseudocode

In this section, we detail the pseudocode of DFRD combined with a PT-based method in Algorithm 1.

## D    Budget Distribution

In our work, in order to simulate model-heterogeneous scenarios and investigate the effect of the different model heterogeneity distributions. We formulate exponentially distributed budgets for a given $N$: $R_i = [\frac{1}{2}]^{\min\{\sigma, \lfloor \frac{\rho \cdot i}{N} \rfloor\}} (i \in [N])$, where $\sigma$ and $\rho$ are both positive integers. Specifically, $\sigma$ controls the smallest capacity model, i.e., $\left[\frac{1}{2}\right]^{\sigma}$-width w.r.t. global model. Also, $\rho$ manipulates client budget distribution. The larger the $\rho$ value, the higher the proportion of the smallest capacity models. We now present client budget distributions under several different settings about parameters $\sigma$ and $\rho$. For example, we fix $\sigma = 4$, that is, the smallest capacity model is $\frac{1}{16}$-width w.r.t. global model. Assuming 10 clients participate in federated learning, we have: For any $i \in [10]$,

$$R_i = [\frac{1}{2}]^{\min\{4, \lfloor \frac{\rho \cdot i}{10} \rfloor\}}. \tag{8}$$

**Algorithm 1** The pseudocode for DFRD combined with PT-based method.

---

**Input:** $T$: communication round; $N$: the number of clients; $S$: the number of sampled active clients per communication round; $B$: batch size. (**client side**) $I_c$: synchronization interval; $\eta_c$: learning rate for SGD. (**server side**) $I$: iteration of the training procedure in server; $(I_g, I_d)$: inner iterations of training the generator and the global model; $(\eta_g, b_1, b_2)$: learning rate and momentums of Adam for the generator; $\eta_d$: learning rate of the global model; $(\beta_{tran}, \beta_{div})$: hyper-parameters in *training generator*; $(\lambda, \alpha)$: control parameters in *robust model distillation*.

1: Initial state $\boldsymbol{\theta}^0$, $\boldsymbol{w}^0$ and $\widetilde{\boldsymbol{w}}^0 = \mathbf{0}$.
2: Initial width capability $\{R_i\}_{i \in [N]}$.
3: Initial weighting counter $\Phi = \{\tau_{i,y}\}_{i \in [N], y \in [C]} \in \mathbb{R}^{N \times C}$, label distribution $P = \{p(y)\}_{y \in [C]} \in \mathbb{R}^C$ and label counter $LC = \{0\}_{i \in [N], y \in [C]} \in \mathbb{R}^{N \times C}$.
4: **for** $t = 0, \cdots, T-1$ **do**
5:     **On Server:**
6:     Sample a subset $\mathcal{S}_t$ with $S$ active clients.
7:     Transmit $\boldsymbol{\theta}_i^t$ (with $R_i$) extracted from $\boldsymbol{\theta}^t$ using sub-model extraction scheme (e.g., *random* [30], *static* [28, 29] or *rolling* [31]) to client $i \in \mathcal{S}_t$.
8:     **On Clients:**
9:     **for** $i \in \mathcal{S}_t$ parallel **do**
10:         $\boldsymbol{\theta}_i^{t+1}, L_i = \text{Client\_Update}(\boldsymbol{\theta}_i^t, \eta_c, I_c, B)$, # see Algorithm 2
11:         Send $\boldsymbol{\theta}_i^{t+1}$ and $L_i$ to server.
12:     **end for**
13:     **On Server:**
14:     Aggregate $\{\boldsymbol{\theta}_i^{t+1}\}_{i \in \mathcal{S}_t}$ according to Eq. (1) and yield $\boldsymbol{\theta}^{t+1}$.
15:     **for** $i \in \mathcal{S}_t$ **do**
16:         $LC[i, :] \leftarrow L_i$.
17:     **end for**
18:     Compute $\Phi$ and $P$ according to $LC$ by using Eq. (9) and (10), respectively.
19:     $\boldsymbol{\theta}^{t+1}, \boldsymbol{w}^{t+1} = \text{Server\_Update}(\{\boldsymbol{\theta}_i^{t+1}\}_{i \in \mathcal{S}_t}, \boldsymbol{\theta}^{t+1}, \boldsymbol{w}^t, \widetilde{\boldsymbol{w}}^t, \Phi, P,$
20:                               $I, I_g, I_d, \eta_g, b_1, b_2, \eta_d, \beta_{tran}, \beta_{div}, \alpha, B)$. # see Algorithm 3
21:     $\widetilde{\boldsymbol{w}}^{t+1} = \lambda \cdot \widetilde{\boldsymbol{w}}^t + (1 - \lambda) \cdot \boldsymbol{w}^{t+1}$.
22:     $LC = \{0\}_{i \in [N], y \in [C]}$.
23: **end for**
**Output:** $\boldsymbol{\theta}^T$

---

- If we set $\rho = 5$, $\{R_i\}_{i \in [10]} = \{1, \frac{1}{2}, \frac{1}{2}, \frac{1}{4}, \frac{1}{4}, \frac{1}{8}, \frac{1}{8}, \frac{1}{16}, \frac{1}{16}, \frac{1}{16}, \frac{1}{16}\}$;

- If we set $\rho = 10$, $\{R_i\}_{i \in [10]} = \{\frac{1}{2}, \frac{1}{4}, \frac{1}{8}, \frac{1}{16}, \frac{1}{16}, \frac{1}{16}, \frac{1}{16}, \frac{1}{16}, \frac{1}{16}, \frac{1}{16}\}$;

- If we set $\rho = 40$, $\{R_i\}_{i \in [10]} = \{\frac{1}{16}, \frac{1}{16}, \frac{1}{16}, \frac{1}{16}, \frac{1}{16}, \frac{1}{16}, \frac{1}{16}, \frac{1}{16}, \frac{1}{16}, \frac{1}{16}\}$.

Note that if $\rho \geq 4N$, each client holds $[\frac{1}{2}]^\sigma$-width w.r.t. global model. More budget distributions can refer to the literature [24, 29, 31].

## E   Complete Experimental Setup

In this section, for the convenience of the reader, we review the experimental setup for all the implemented algorithms on top of FMNIST, SVHN, CIFAR-10, CIFAR-100, Tiny-ImageNet and FOOD101.

**Datasets.** In this paper, we evaluate different methods with six real-world image classification task-related datasets, namely Fashion-MNIST [69] (FMNIST in short), SVHN [70], CIFAR-10, CIFAR-100 [71], Tiny-imageNet[5] and Food101 [73]. We detail the six datasets in Appendix B. To simulate data heterogeneity across clients, as in previous works [34, 37, 38], we use Dirichlet process $Dir(\omega)$ to partition the training set for each dataset, thereby allocating local training data for each

---

[5]http://cs231n.stanford.edu/tiny-imagenet-200.zip

---

**Algorithm 2** Client_Update($\boldsymbol{\theta}$, $\eta$, $I$, $B$) # Take client $i$ as an example

---

1: Set $\boldsymbol{\theta}_i = \boldsymbol{\theta}$ and $L_i = \{0\}_{y \in [C]} \in \mathbb{R}^C$.
2: Cache data $Cache = []$.
3: **for** $e = 0, 1, \ldots, I - 1$ **do**
4:      Sample $\mathcal{B} = \{(\boldsymbol{x}_i^b, y_i^b)\}_{b=1}^B$ from $\{(\boldsymbol{X}_i, \boldsymbol{Y}_i)\}$.
5:      $\boldsymbol{\theta}_i \leftarrow \boldsymbol{\theta}_i - \frac{\eta}{B} \sum_{b \in [B]} \nabla \mathcal{L}_i(f_i(\boldsymbol{x}_i^b, \boldsymbol{\theta}_i), y_i^b)$, where $\mathcal{L}_i$ denotes the cross-entropy loss.
6:      **for** $(\boldsymbol{x}, y)$ in $\mathcal{B}^e$ **do**
7:          **if** $(\boldsymbol{x}, y)$ not in $Cache$ **then**
8:              $Cache \leftarrow Cache \cup \{(\boldsymbol{x}, y)\}$.
9:          **end if**
10:      **end for**
11:      **for** $(\boldsymbol{x}, y)$ in $Cache$ **do**
12:          $L_i^y \leftarrow L_i^y + 1$, where $L_i^y$ denotes the value of the $y^{th}$ element in $L_i$.
13:      **end for**
14: **end for**
15: **Output:** $\boldsymbol{\theta}_i, L_i$.

---

**Algorithm 3** Server_Update($\{\boldsymbol{\theta}_i\}_{i \in \mathcal{S}_t}$, $\boldsymbol{\theta}$, $\boldsymbol{w}$, $\widetilde{\boldsymbol{w}}$, $\Theta$, $P$, $I$, $I_g$, $I_d$, $\eta_g$, $b_1$, $b_2$, $\eta_d$, $\beta_{tran}$, $\beta_{div}$, $\alpha^\star$, $B$)

---

1: **for** $e = 0, 1, \ldots, I - 1$ **do**
2:      Sample a batch of $\{\boldsymbol{z}^b, y^b\}_{b=1}^B$, where $\boldsymbol{z} \sim \mathcal{N}(\boldsymbol{0}, \boldsymbol{I})$ and $y \sim p(y) := P[y]$.
3:      Set $\boldsymbol{m} = 0$ and $\boldsymbol{v} = 0$.
4:      **for** $e_g = 0, 1, \ldots, I_g - 1$ **do**
5:          Generate $\{\boldsymbol{s}^b\}_{b=1}^B$ with $\{\boldsymbol{z}^b, y^b\}_{b=1}^B$ and $G(\cdot)$,
6:          Compute generator loss $\mathcal{L}_{gen} = \mathcal{L}_{fid} + \beta_{tran} \cdot \mathcal{L}_{tran} + \beta_{div} \cdot \mathcal{L}_{div}$
7:                                                  by using $\Theta$ and Eq. (2)-(5),
8:          $\boldsymbol{g} = \frac{1}{B} \sum_{b \in [B]} \nabla_{\boldsymbol{w}} \mathcal{L}_{gen}$,
9:          $\boldsymbol{m} \leftarrow b_1 \cdot \boldsymbol{m} + (1 - b_1) \cdot \boldsymbol{g}$, $\boldsymbol{v} \leftarrow b_2 \cdot \boldsymbol{v} + (1 - b_2) \cdot \boldsymbol{g}^2$,
10:          $\hat{\boldsymbol{m}} \leftarrow \boldsymbol{m}/(1 - b_1)$, $\hat{\boldsymbol{v}} \leftarrow \boldsymbol{v}/(1 - b_2)$,
11:          $\boldsymbol{w} \leftarrow \boldsymbol{w} - \eta_g \hat{\boldsymbol{m}}/(\sqrt{\hat{\boldsymbol{v}}} + 10^{-8})$.
12:      **end for**
13:      **for** $e_d = 0, 1, \ldots, I_d - 1$ **do**
14:          Generate $\{\boldsymbol{s}^b\}_{b=1}^B$ with $\{\boldsymbol{z}^b, y^b\}_{b=1}^B$ and $G(\cdot)$.
15:          **if** $\widetilde{\boldsymbol{w}} \neq \boldsymbol{0}$ **then**
16:              Sample a batch of $\{\widetilde{\boldsymbol{z}}^b, \widetilde{y}^b\}_{b=1}^B$, where $\widetilde{\boldsymbol{z}} \sim \mathcal{N}(\boldsymbol{0}, \boldsymbol{I})$ and $\widetilde{y} \sim p(\widetilde{y}) := P[\widetilde{y}]$,
17:              Generate $\{\widetilde{\boldsymbol{s}}^b\}_{b=1}^B$ with $\{\widetilde{\boldsymbol{z}}^b, \widetilde{y}^b\}_{b=1}^B$ and $\widetilde{G}(\cdot)$,
18:              Compute global model loss $\mathcal{L}_{md}$ by using $\Theta$ and Eq. (7) with $\alpha = \alpha^\star$,
19:              $\boldsymbol{\theta} \leftarrow \boldsymbol{\theta} - \frac{\eta_d}{B} \sum_{b \in [B]} \nabla_{\boldsymbol{\theta}} \mathcal{L}_{md}$.
20:          **else**
21:              Compute global model loss $\mathcal{L}_{kl}$ by using $\Theta$ and Eq. (7) with $\alpha = 0$,
22:              $\boldsymbol{\theta} \leftarrow \boldsymbol{\theta} - \frac{\eta_d}{B} \sum_{b \in [B]} \nabla_{\boldsymbol{\theta}} \mathcal{L}_{kl}$.
23:          **end if**
24:      **end for**
25: **end for**
26: **Output:** $\boldsymbol{\theta}, \boldsymbol{w}$.

---

client. It is worth noting that $\omega$ is the concentration parameter and smaller $\omega$ corresponds to stronger data heterogeneity.

**Baselines.** We compare DFRD to FedFTG [48] and DENSE [49], which are the most relevant methods to our work. To verify the superiority of DFRD, on the one hand, DFRD, FedFTG and DENSE are directly adopted on the server to transfer the knowledge of the local models to a randomly initialized global model. We call them collectively **data-free methods**. On the other hand, they are utilized as **fine-tuning methods** to improve the global model's performance after computing weighted average per communication round. In this case, in each communication round, the preliminary global model

is obtained using FedAvg [14] in FL with homogeneous models, whereas in FL with heterogeneous models, the PT-based methods, including HeteroFL [29], Federated Dropout [30] (FedDp for short) and FedRolex [31], are employed to get the preliminary global model. [6]

**Configurations.** Unless otherwise stated, all experiments are performed on a centralized network with $N = 10$ active clients. We set $\omega \in \{0.01, 0.1, 1.0\}$ to mimic different data heterogeneity scenarios. A visualization of the data partitions for the six datasets at varying $\omega$ values can be found in Fig. 6 and 7. To simulate model-heterogeneous scenarios, we formulate exponentially distributed budgets for a given $N$: $R_i = [\frac{1}{2}]^{\min\{\sigma, \lfloor \frac{\rho \cdot i}{N} \rfloor\}} (i \in [N])$, where $\sigma$ and $\rho$ are both positive integers. We fix $\sigma = 4$ and consider $\rho \in \{5, 10, 40\}$. See Appendix D for more details. Unless otherwise specified, we set $\beta_{tran}$ and $\beta_{div}$ both to 1 in *training generator*, while in *robust model distillation*, we set $\lambda = 0.5$ and $\alpha = 0.5$. For each client, we fix synchronization interval $I_c = 20$ and select $\eta_c$ from $\{0.001, 0.01, 0.1\}$. For fairness, the popular SGD procedure is employed to perform local update steps for each client. On the server, we take the values 10, 5 and 2 for $I$, $I_g$ and $I_d$ respectively. Also, SGD and Adam are applied to optimize the global model and generator, respectively. The learning rate $\eta_d$ for SGD is searched over the range of $\{0.001, 0.01, 0.1\}$ and the best one is picked. We set $\eta_g = 0.0002$, $b_1 = 0.5$ and $b_2 = 0.999$ for Adam. For all update steps, we set batch size $B$ to $64$.

To verify the superiority of DFRD, we execute substantial comparative experiments in terms of data and model heterogeneity for FL, respectively.

Firstly, we investigate the performance of DFRD against that of baselines on FMNIST, SVHN, CIFAR-10 and CIFAR-100 w.r.t. different levels of data heterogeneity in FL with homogeneous models. To be specific, For FMNIST, SVHN and CIFAR-10, a four-layer convolutional neural network with BatchNorm is implemented for each client. We fix the total communication rounds to 100, i.e., $T = 100$. For CIFAR-100, each client implements a ResNet18 [72] architecture. We fix the total communication rounds to 500, i.e., $T = 500$.

Then, we look into the performance of DFRD against that of baselines on SVHN, CIFAR-10, Tiny-ImageNet and FOOD101 in terms of different levels of model heterogeneity distribution in FL. Concretely, for SVHN (CIFAR-10), we set concentration parameter $\omega = 0.1$. We equip the server with a four-layer convolutional neural network with BatchNorm as the global model, and extract suitable sub-models from the global model according to the width capability of each client. We fix the total communication rounds to 200 (300), i.e., $T = 200$ (300). For Tiny-ImageNet (FOOD101), we set the concentration parameter $\omega = 1.0$. Meanwhile, a ResNet20 [72] architecture is equipped on the server as the global model, and appropriate sub-models are extracted from the global model according to the width capability of each client. We fix the total communication rounds to 600 (600), i.e., $T = 600$ (600). In addition, it is crucial to equip each dataset with a suitable generator on the server. The details of generator frameworks are shown in Tables 7-9. Moreover, for FMNIST, SVHN and CIFAR-10, we set the random noise dimension $d = 64$. For CIFAR-100, Tiny-ImageNet and FOOD101, we set the random noise dimension $d = 100$. All baselines leverage the same setting as ours.

Table 7: Generator for FMNIST

| $\boldsymbol{z} \in \mathbb{R}^d \sim \mathcal{N}(\boldsymbol{0}, \boldsymbol{I}), y \in [C]$ |
| :---: |
| $\boldsymbol{h} = o(\boldsymbol{z}, y) \rightarrow d$ |
| Reshape($\boldsymbol{h}$)$\rightarrow d \times 1 \times 1$ |
| Relu(ConvTransposed2d($d$, 512)) $\rightarrow B \times 512 \times 4 \times 4$ |
| Relu(BN(ConvTransposed2d(512, 256))) $\rightarrow 256 \times 8 \times 8$ |
| Relu(BN(ConvTransposed2d(256, 128))) $\rightarrow 128 \times 16 \times 16$ |
| Tanh(ConvTransposed2d(128, 1)) $\rightarrow 1 \times 32 \times 32$ |

**Computing devices and platforms:**

- OS: Ubuntu 20.04.3 LTS

---

[6]As an example, DFRD+FedRelox indicates that DFRD is used as a fine-tuning method to improve the performance of FedRelox, and others are similar. See Tables 1 and 2 for details.

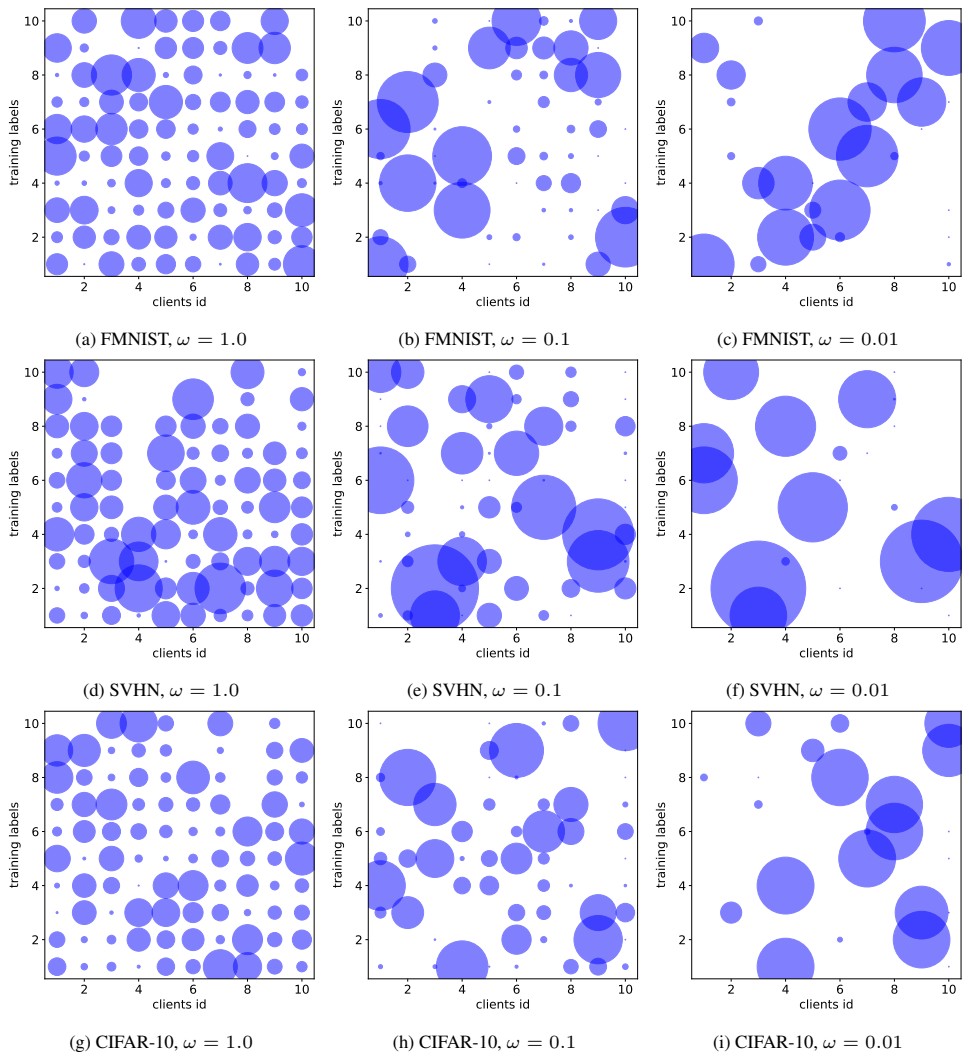

Figure 6: Data heterogeneity among clients is visualized on three datasets (FMNIST, SVHN, CIFAR-10), where the $x$-axis represents the clients id, the $y$-axis represents the class labels on the training set, and the size of scattered points represents the number of training samples with available labels for that client.

- CPU: AMD EPYC 7763 64-Core Processor
- CPU Memory: 2 T
- GPU: NVIDIA Tesla A100 PCIe
- GPU Memory: 80GB
- Programming platform: Python 3.8.16
- Deep learning platform: PyTorch 1.12.1

## F  Dynamic Weighting and Label Sampling

Here, we study how to determine $\tau_{i,y}$ and $p(y)$. We first consider $\tau_{i,y}$. Existing works either assign the same weights to the logits outputs of different local models, i.e., $\tau_{i,y} = \frac{1}{|\mathcal{S}_t|}$ [40, 47, 49] (▲), or assign weights to them based on the respective static data distribution of the clients, i.e., $\tau_{i,y} = n_i^y / n_{\mathcal{S}_t}^y$, where $n_{\mathcal{S}_t}^y = \sum_{j \in [\mathcal{S}_t]} n_j^y$ and $n_i^y$ denotes the number of data points with label $y$ on the $i$-th client [48](▼). For the former, since the importance of knowledge is different among local

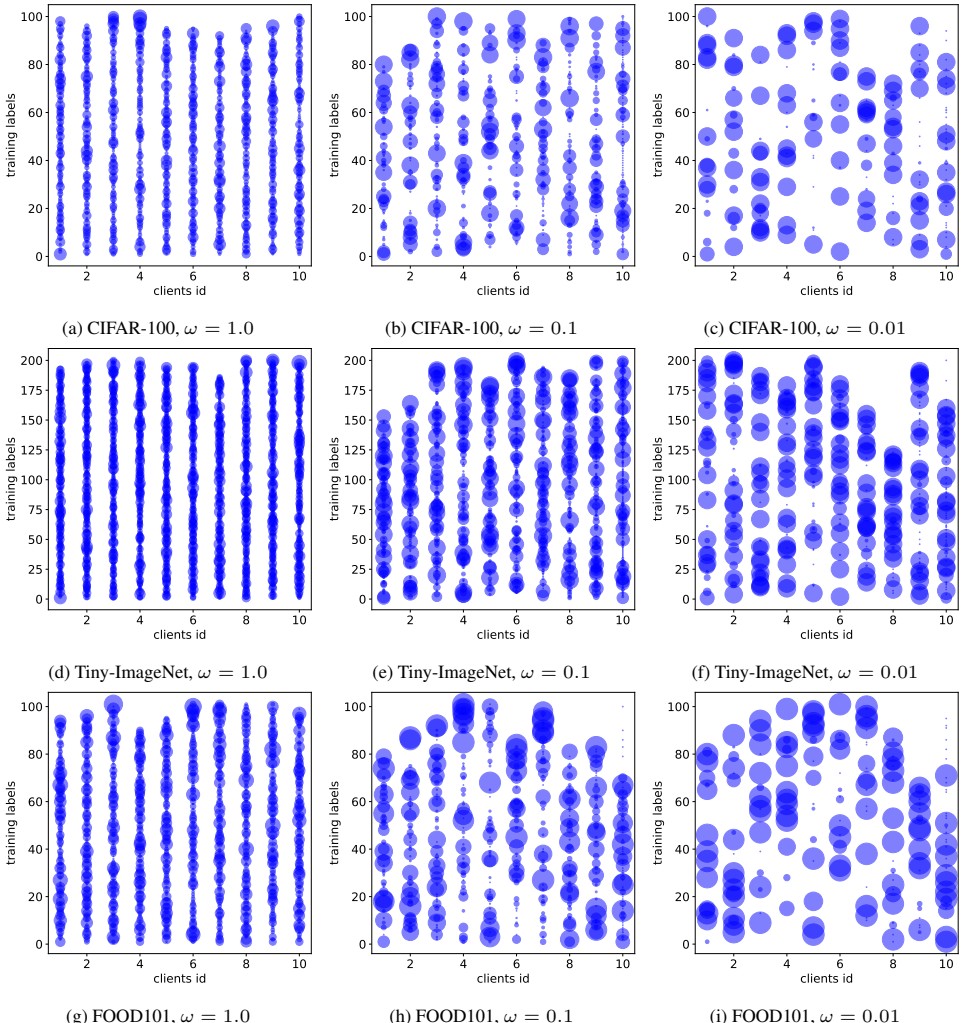

Figure 7: Data heterogeneity among clients is visualized on three datasets (CIFAR-100, Tiny-ImageNet and FOOD101), where the $x$-axis represents the clients id, the $y$-axis represents the class labels on the training set, and the size of scattered points represents the number of training samples with available labels for that client.

models in FL with label distribution skewness (i.e., data heterogeneity), the important knowledge cannot be correctly identified and utilized. The latter, meanwhile, may not be feasible in practical FL, where on the one hand the label distribution on each client may be constantly changing [65–68], and on the other hand, there may exist labels held by clients not involved in training under batch training. Therefore, in FL with heterogeneous data, neither ▲ nor ▼ can reasonably control the weight of the logits outputs among local models and yield a biased ensemble model (i.e., $\sum_{i \in \mathcal{S}_t} \tau_{i,y} f_i(\boldsymbol{s}, \boldsymbol{\theta}_i)$ in Eq. (2), (3) and (7)), thereby misleading the training of generator and global model.

To overcome the mentioned problems, we propose dynamic weighting, which only calculates the proportion of data involved in the local update in $\mathcal{S}_t$, i.e.,

$$\tau_{i,y} = n_{i,t}^y / n_{\mathcal{S}_t,t}^y, \tag{9}$$

where $n_{\mathcal{S}_t,t}^y = \sum_{j \in [\mathcal{S}_t]} n_{j,t}^y$ and $n_{i,t}^y (\leq n_i^y)$ denotes the number of data with label $y$ involved in training on client $i$ at round $t$ (■). If $n_{i,t}^y = 0$, it means the data with label $y$ held by client $i$ is not involved in training at $t$ round, or client $i$ has no data with label $y$.

Table 8: Generator for SVHN, CIFAR-10 and CIFAR-100

| |
|---|
| $\boldsymbol{z} \in \mathbb{R}^d \sim \mathcal{N}(\mathbf{0}, \boldsymbol{I}), y \in [C]$ |
| $\boldsymbol{h} = o(\boldsymbol{z}, y) \rightarrow d$ |
| Reshape($\boldsymbol{h}$)$\rightarrow d \times 1 \times 1$ |
| Relu(ConvTransposed2d($d$, 512)) $\rightarrow B \times 512 \times 4 \times 4$ |
| Relu(BN(ConvTransposed2d(512, 256))) $\rightarrow 256 \times 8 \times 8$ |
| Relu(BN(ConvTransposed2d(256, 128))) $\rightarrow 128 \times 16 \times 16$ |
| Tanh(ConvTransposed2d(128, 3)) $\rightarrow 3 \times 32 \times 32$ |

Table 9: Generator for Tiny-ImageNet and FOOD101

| |
|---|
| $\boldsymbol{z} \in \mathbb{R}^d \sim \mathcal{N}(\mathbf{0}, \boldsymbol{I}), y \in [C]$ |
| $\boldsymbol{h} = o(\boldsymbol{z}, y) \rightarrow d$ |
| Reshape($\boldsymbol{h}$)$\rightarrow d \times 1 \times 1$ |
| Relu(ConvTransposed2d($d$, 512)) $\rightarrow B \times 512 \times 4 \times 4$ |
| Relu(BN(ConvTransposed2d(512, 256))) $\rightarrow 256 \times 8 \times 8$ |
| Relu(BN(ConvTransposed2d(256, 128))) $\rightarrow 128 \times 16 \times 16$ |
| Relu(BN(ConvTransposed2d(128, 64))) $\rightarrow 64 \times 32 \times 32$ |
| Tanh(ConvTransposed2d(64, 3)) $\rightarrow 3 \times 64 \times 64$ |

We now give the label distribution $p(y)$. Typically, $p(y)$ correlates with $\tau_{i,y}$. For $\blacksquare$,

$$p(y) = n_{\mathcal{S}_t,t}^y / \sum_{y \in [C]} n_{\mathcal{S}_t,t}^y. \tag{10}$$

Similarly, $p(y) = \frac{1}{C}$ for $\blacktriangle$ and $p(y) = n_{\mathcal{S}_t}^y / \sum_{y \in [C]} n_{\mathcal{S}_t}^y$ for $\blacktriangledown$. [7] Intuitively, $\tau_{i,y}$ and $p(y)$ (i.e. $\blacksquare$) obtained from the label distribution of clients participating in local training ensure that the generator accurately approximates the data distribution from the labels that they know best.

To verify the utility of $\blacktriangle$, $\blacktriangledown$ and $\blacksquare$, we conduct experiments on SVHN, CIFAR-10 and CIFAR-100, and the test results are presented in Table 10. Note that the experimental setup here is consistent with Section 4.3 in the main paper and we do not repeat it. From Table 10, we can see that $\blacksquare$ significantly outperforms $\blacktriangle$ and $\blacktriangledown$ in terms of global test accuracy, which indicates that $\blacksquare$ is more effective in aggregating logits outputs of local models. We attribute the merits of the proposed $\blacksquare$ to two things. Firstly, in the local update phase, $\blacksquare$ only needs to compute the real data with labels involved in the training, which allows DFRD to effectively overcome the drift among local models, and thus enable a compliant global training space. Secondly, the label distribution derived from $\blacksquare$ generates labels that participate in local updates on a probabilistic basis, so as to avoid misleading the training of the conditional generator $G(\cdot)$.

## G    Impacts of the number of clients and stragglers

In real-world FL scenarios with both data and model heterogeneity, the number of clients participating in the federation may be large [5]. Also, due to differences in available modes among clients, some clients may abort the training midway (i.e., stragglers) [34]. Therefore, in this section, we evaluate the test performance of FedReloX (as a baseline) and FedReloX+DFRD with varying numbers of clients and stragglers on SVHN and CIFAR-10. For the number of clients $N$, we select $N$ from $\{10, 20, 50, 100, 200\}$. For stragglers, we consider its opposite, that is, the number of active clients

---

[7]Note that $\blacktriangle$, $\blacktriangledown$ and $\blacksquare$ denote the average, static and dynamic weighting schemes as well as the corresponding label sampling schemes, respectively.

Table 10: Test accuracy (%) comparison among ▲, ▼ and ■ over SVHN, CIFAR-10/100.

| W. S. | SVHN | CIFAR-10 | CIFAR-100 |
|---|---|---|---|
| ▲ | $30.56_{\pm14.34}$ $(15.93_{\pm1.42})$ | $21.61_{\pm1.71}$ $(15.77_{\pm0.96})$ | $24.98_{\pm0.99}$ $(12.19_{\pm0.40})$ |
| ▼ | $31.40_{\pm12.72}$ $(15.89_{\pm1.35})$ | $22.80_{\pm1.34}$ $(16.29_{\pm1.43})$ | $26.75_{\pm2.00}$ $(12.66_{\pm0.56})$ |
| ■ | $\mathbf{34.78_{\pm9.19}}$ $(\mathbf{15.99_{\pm1.53}})$ | $\mathbf{25.57_{\pm1.37}}$ $(\mathbf{16.74_{\pm0.72}})$ | $\mathbf{28.08_{\pm0.94}}$ $(\mathbf{13.03_{\pm0.16}})$ |

participating in training per communication round. Specifically, we fix $N = 100$ and set the number of active clients $S \in \{5, 10, 50, 100\}$. Moreover, to ensure consistency, we set the width capability of each client to $\frac{1}{16}$ when $N$ is given.

Table 11 shows the test performances among different $N$. We can observe that as $N$ increases, the global test accuracy of FedRolex+DFRD decreases uniformly on SVHN, while it increases and then decreases on CIFAR-10. For the latter, this seems inconsistent with the observations in [14, 49]. This is an interesting result. We conjecture that for a given task, increasing $N$ within a certain range, FedRolex+DFRD may enhance the global model, but the performance of the global model still degrades if a threshold is exceeded. Besides, FedRolex+DFRD significantly outperforms FedRolex for each given $N$ in terms of global test accuracy, which improves by an average of $9.75\%$ on SVHN and $8.01\%$ on CIFAR-10. This shows that DFRD can effectively augment the global model for different $N$ in FL with both data and model heterogeneity.

Table 12 reports the test results for different $S$. A higher $S$ means more active clients upload model per communication round. Note that due to the existence of stragglers, the number of rounds and times of training among clients may not be identical. For brevity, we only display the global test accuracy. We can clearly see that the performance of FedRolex and FedRolex+DFRD improves uniformly with increasing $S$, where FedRolex+DFRD consistently dominates FedRolex in terms of global test accuracy. This means that stragglers are detrimental to the global model. As such, stragglers pose new challenges for data-free knowledge distillation, which we leave for future work.

Table 11: Test accuracy (%) over different $N$ on SVHN and CIFAR-10.

| Dataset | SVHN | | CIFAR-10 | |
|---|---|---|---|---|
| $N$ | FedRolex | FedRolex+DFRD | FedRolex | FedRolex+DFRD |
| 10 | $22.39_{\pm15.28}$ $(13.59_{\pm2.11})$ | $\mathbf{32.86_{\pm15.54}}$ $(\mathbf{15.17_{\pm0.66}})$ | $14.37_{\pm2.91}$ $(17.11_{\pm1.29})$ | $\mathbf{19.86_{\pm2.76}}$ $(\mathbf{17.77_{\pm1.16}})$ |
| 20 | $18.20_{\pm1.43}$ $(12.48_{\pm0.47})$ | $\mathbf{31.32_{\pm10.16}}$ $(\mathbf{13.69_{\pm1.38}})$ | $13.60_{\pm0.68}$ $(16.30_{\pm0.98})$ | $\mathbf{20.20_{\pm1.30}}$ $(\mathbf{17.11_{\pm0.92}})$ |
| 50 | $16.83_{\pm1.02}$ $(11.73_{\pm0.45})$ | $\mathbf{27.62_{\pm3.41}}$ $(\mathbf{12.48_{\pm0.90}})$ | $12.75_{\pm0.71}$ $(16.50_{\pm1.14})$ | $\mathbf{21.14_{\pm2.17}}$ $(\mathbf{16.53_{\pm0.78}})$ |
| 100 | $15.38_{\pm1.22}$ $(10.80_{\pm0.14})$ | $\mathbf{24.22_{\pm4.75}}$ $(\mathbf{11.56_{\pm0.53}})$ | $12.25_{\pm1.39}$ $(16.26_{\pm1.01})$ | $\mathbf{23.68_{\pm1.25}}$ $(\mathbf{16.88_{\pm0.72}})$ |
| 200 | $10.33_{\pm0.17}$ $(9.45_{\pm1.59})$ | $\mathbf{15.85_{\pm2.64}}$ $(\mathbf{10.01_{\pm0.21}})$ | $10.14_{\pm2.11}$ $(14.65_{\pm0.99})$ | $\mathbf{18.27_{\pm0.56}}$ $(\mathbf{15.44_{\pm0.53}})$ |

# H  Full learning curves.

In this section, we report the complete learning curves corresponding to Tables 1 and 2 in the main paper, see Figures 8 to 12.

Table 12: Test accuracy (%) over different $S$ per communication round on SVHN and CIFAR-10.

| Dataset | SVHN | | CIFAR-10 | |
|---|---|---|---|---|
| $S$ | FedRolex | FedRolex+DFRD | FedRolex | FedRolex+DFRD |
| 5 | $13.86_{\pm 2.06}$ | $\mathbf{17.58_{\pm 0.47}}$ | $10.22_{\pm 0.46}$ | $\mathbf{18.59_{\pm 0.44}}$ |
| 10 | $14.64_{\pm 1.17}$ | $\mathbf{19.29_{\pm 0.17}}$ | $11.67_{\pm 1.34}$ | $\mathbf{21.22_{\pm 0.08}}$ |
| 50 | $15.17_{\pm 1.63}$ | $\mathbf{22.85_{\pm 4.23}}$ | $12.73_{\pm 0.71}$ | $\mathbf{23.18_{\pm 0.31}}$ |
| 100 | $15.38_{\pm 1.22}$ | $\mathbf{24.22_{\pm 4.75}}$ | $12.79_{\pm 0.71}$ | $\mathbf{23.68_{\pm 1.25}}$ |

# I   Visualization of synthetic data

In this section, we visualize the synthetic data generated by $G(\cdot)$ and $\tilde{G}(\cdot)$ on SVHN, CIFAR-10 and CIFAR-100 based on different merge operators, as shown in Figures 14 to 18. Additionally, we visualize the partial original data of SVHN, CIFAR-10 and CIFAR-100, see Fig. 13 for details. From Figures 13 to 18, it can be observed that all the synthetic data are visually very dissimilar to the original data, which effectively avoids the leakage of the original data. As we state in Section 3.1 of the main paper, a well-trained generator should satisfy several key characteristics: *fidelity*, *transferability*, and *diversity*. Therefore, we analyze the performance of $G(\cdot)$ and $\tilde{G}(\cdot)$ with different merge operators in terms of these three aspects. Firstly, we can easily see from Figures 14 to 18 that $G(\cdot)$ and $\tilde{G}(\cdot)$ based on $mul$, $add$, $cat$ and $ncat$ can capture the shared information of the original data while ensuring data privacy. However, synthetic data generated by $G(\cdot)$ and $\tilde{G}(\cdot)$ based on $none$ (i.e. not conditional generator) is not visually discernible as to whether it extracts valid information from the original data. From the test accuracies reported in Table 4 in the main paper, we can see that DFRD based on $none$ consistently underperforms other competitors in terms of global test accuracy, which indicates that the generator and EMA generator based on $none$ generate lower quality synthetic data. Also, $mul$ consistently outperforms other merge operators w.r.t. global test accuracy, suggesting that the generator and EMA generator based on $mul$ can generate more effective synthetic data. Furthermore, in terms of diversity, the synthetic data generated by $\tilde{G}(\cdot)$ based on $mul$ is visually diverse both inter and intra classes, which suggests that no model collapse occurred during training of the EMA generator. It is worth noting that the synthetic data generated by $\tilde{G}(\cdot)$ is different from that generated by $G(\cdot)$, meaning that the EMA generator can be used as a complement to the generator in DFRD.

# J   Limitations

In the field of Federated Learning (FL), there are many trade-offs, including utility, privacy protection, computational efficiency and communication cost, etc. It is well known that trying to develop a universal FL method that can address all problems is extremely challenging. In this work, we mainly focus on improving the utility of the global model in FL. Next, we discuss some of the limitations of DFRD.

**Privacy Protection.** We acknowledge that it is difficult for DFRD to strictly guarantee privacy without privacy protection. Since DFRD generates synthetic data in the server that is similar to clients' training data, it may violate the privacy regulations in FL. However, according to our observation, DFRD can only capture the shared attributes of the dataset without access to real data, and it is difficult to concretely and visually reveal the unique attributes of individual data (see Figures 14 to 17). In addition, DFRD requires clients to upload the label statistics of the data, which also is at risk of compromising privacy. However, based on our proposed dynamic weighting and label sampling, each client only needs to count the labels of the data involved in local training instead of counting that of the entire local training dataset, which mitigates the privacy leakage to some extent. Although our work does not address the privacy issue, in tasks with high privacy protection requirements, DFRD can incorporate some privacy protection techniques to enhance the reliability of FL systems in practice, such as differential privacy (DP) [74–76]. Note that the combination of DFRD and encryption technologies [77, 78] may not be feasible, since *training generator* and *robust model distillation* in DFRD require knowledge of the structure of the local models. In addition, we argue that the **theoretical guarantee** for privacy protection is crucial. However, It's worth noting that even

in exist well-known efforts [48, 49, 53, 59], including those on handling data and model heterogeneity for FL with the help DFKD [48, 49], comprehensive theoretical analysis concerning the boundaries of data generation is often absent. Given the lack of suitable theoretical frameworks, we concentrated on robust empirical validation, showcasing our method (DFRD). Our results, we believe, robustly demonstrate our method's utility. We intend to delve deeper into theoretical aspects in future work.

**Computational Efficiency and Communication Cost.** To achieve a robust global model, DFRD requires additional computation. Specifically, compared with FedAvg or PT-based methods, the training time of DFRD as a fine-tuning method will be longer, as it needs to additionally train the generator and the global model on the server. In our experiments, DFRD takes two to three times longer to run per communication round than they do. Moreover, compared to FedAvg or PT-based methods, DFRD requires an additional vector of label statistics to be transmitted (see line 11 in Algorithm 1). However, the communication cost of this vector is negligible compared to that of the model.

# K   Broader Impacts

We work on how to train a robust global model with the help of data-free knowledge distillation (DFKD) in the Federated Learning (FL) scenario where data and models are simultaneously heterogeneous. Our work points out that existing methods combining FL and DFKD do not thoroughly study the training of the generator, and neglect the catastrophic forgetting of the global model caused by large distribution shifts of the generator in the scenarios of coexisting data and model heterogeneity. Our proposed DFRD can deal with the said issues well. DFRD exemplifies potential positive impacts on society, enabling a global model with superior performance while ensuring basic privacy protection in real-world FL applications. Meanwhile, DFRD may have negative social impacts related to sensitive information and high resource consumption. While DFRD does not compromise the private information of individual data, it may expose shared information with utility for the performance of the generator. In addition, DFRD-based FL systems require more server-side power resources to train the global model. DFRD does not involve social ethics.

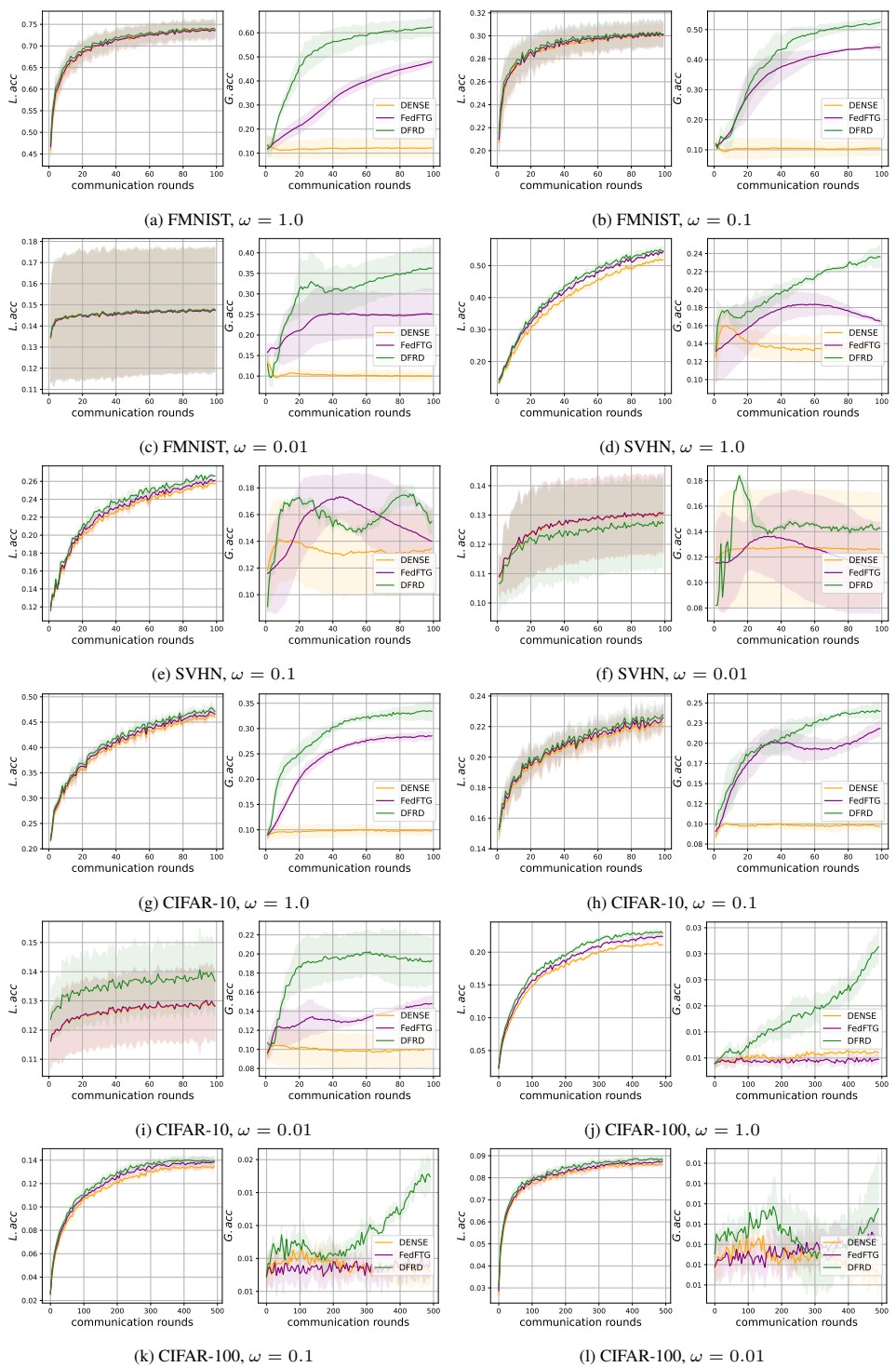

Figure 8: Learning curves of distinct data-free methods across $\omega \in \{0.01, 0.1, 1.0\}$ on FMNIST, SVHN, CIFAR-10 and CIFAR-100.

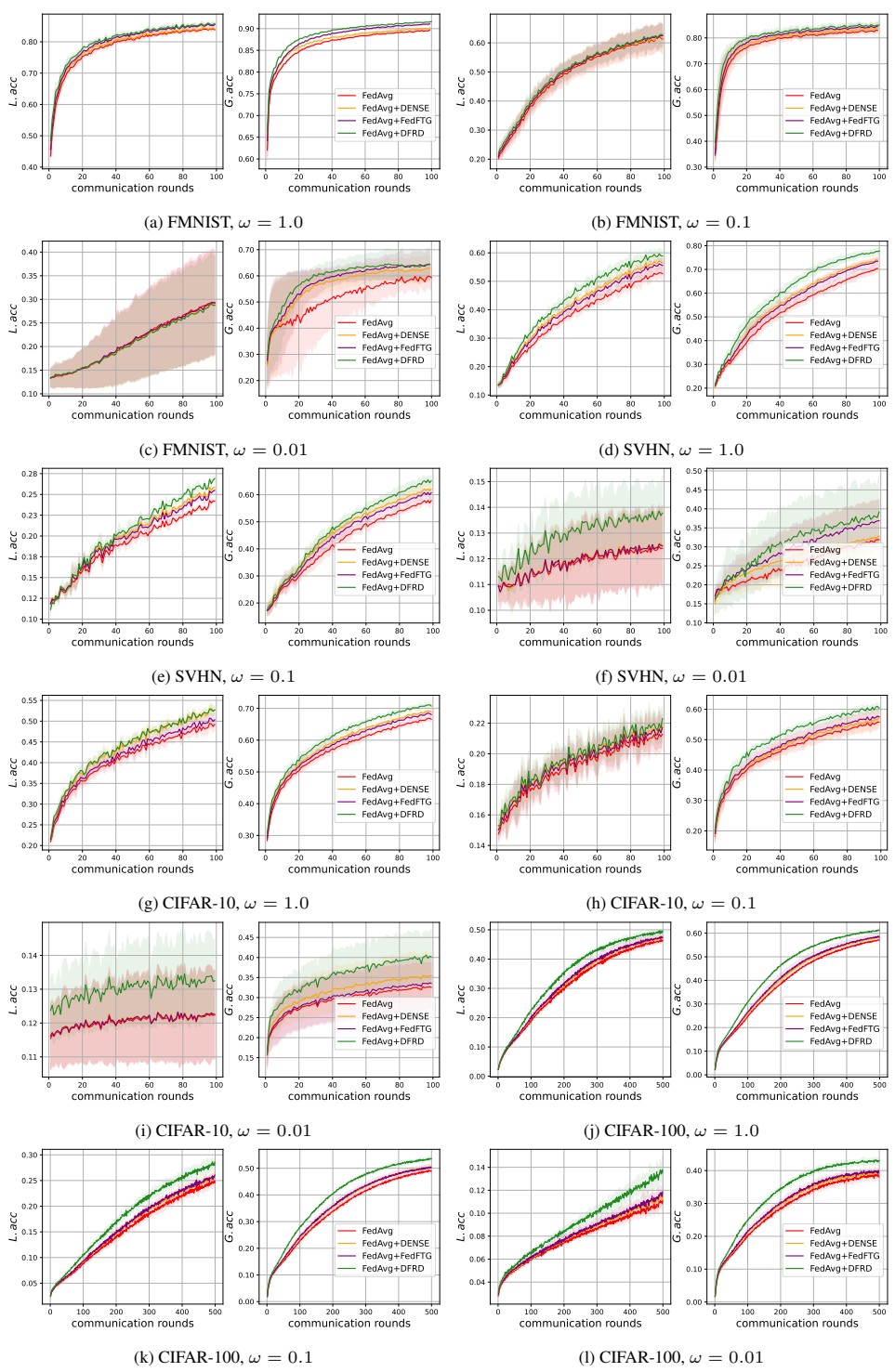

Figure 9: Learning curves of distinct fine-tuning methods based on FedAvg across $\omega \in \{0.01, 0.1, 1.0\}$ on FMNIST, SVHN, CIFAR-10 and CIFAR-100.

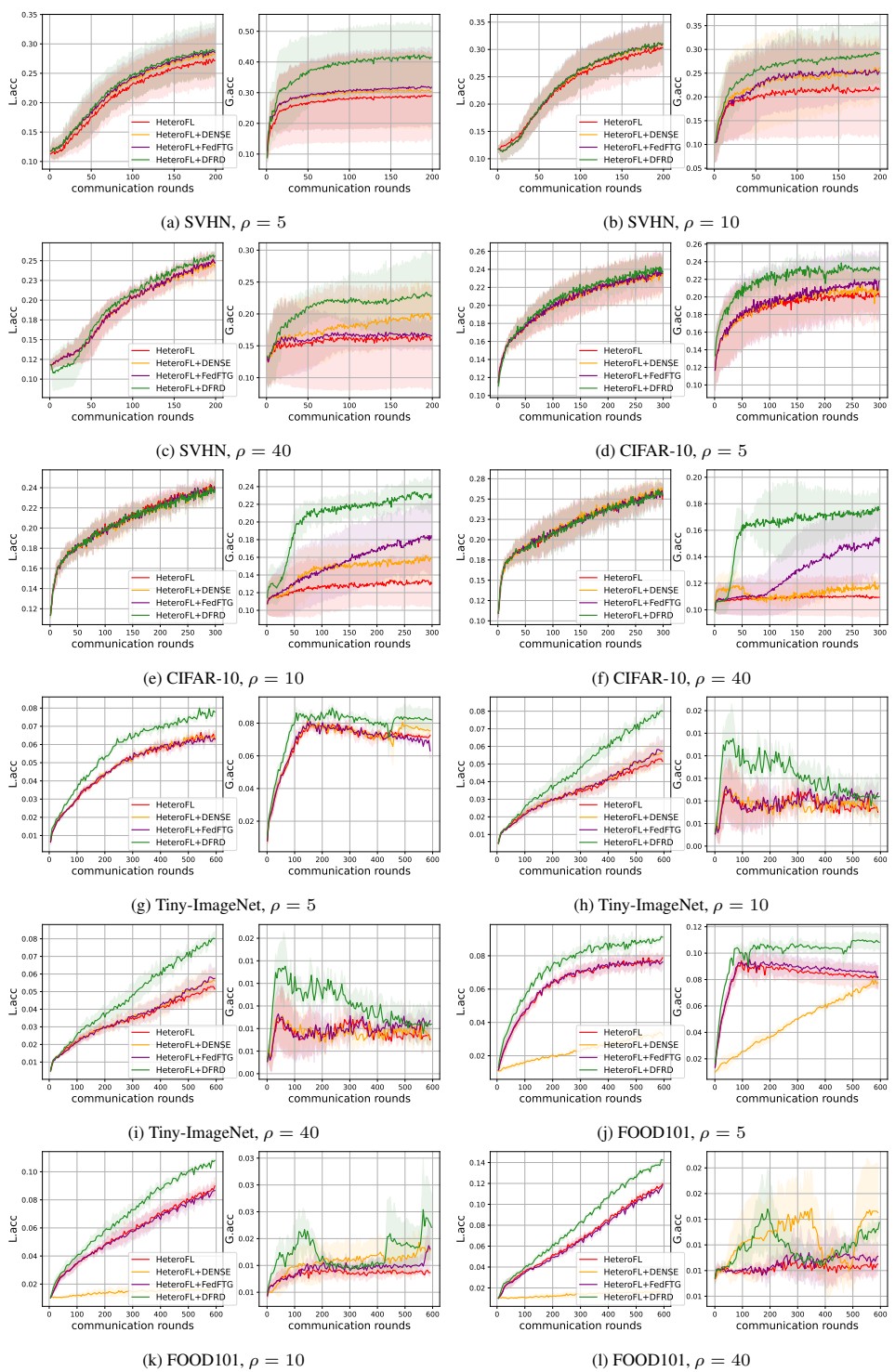

Figure 10: Learning curves of distinct fine-tuning methods based on HeteroFL across $\rho \in \{5, 10, 40\}$ on SVHN, CIFAR-10, Tiny-ImageNet and FOOD101.

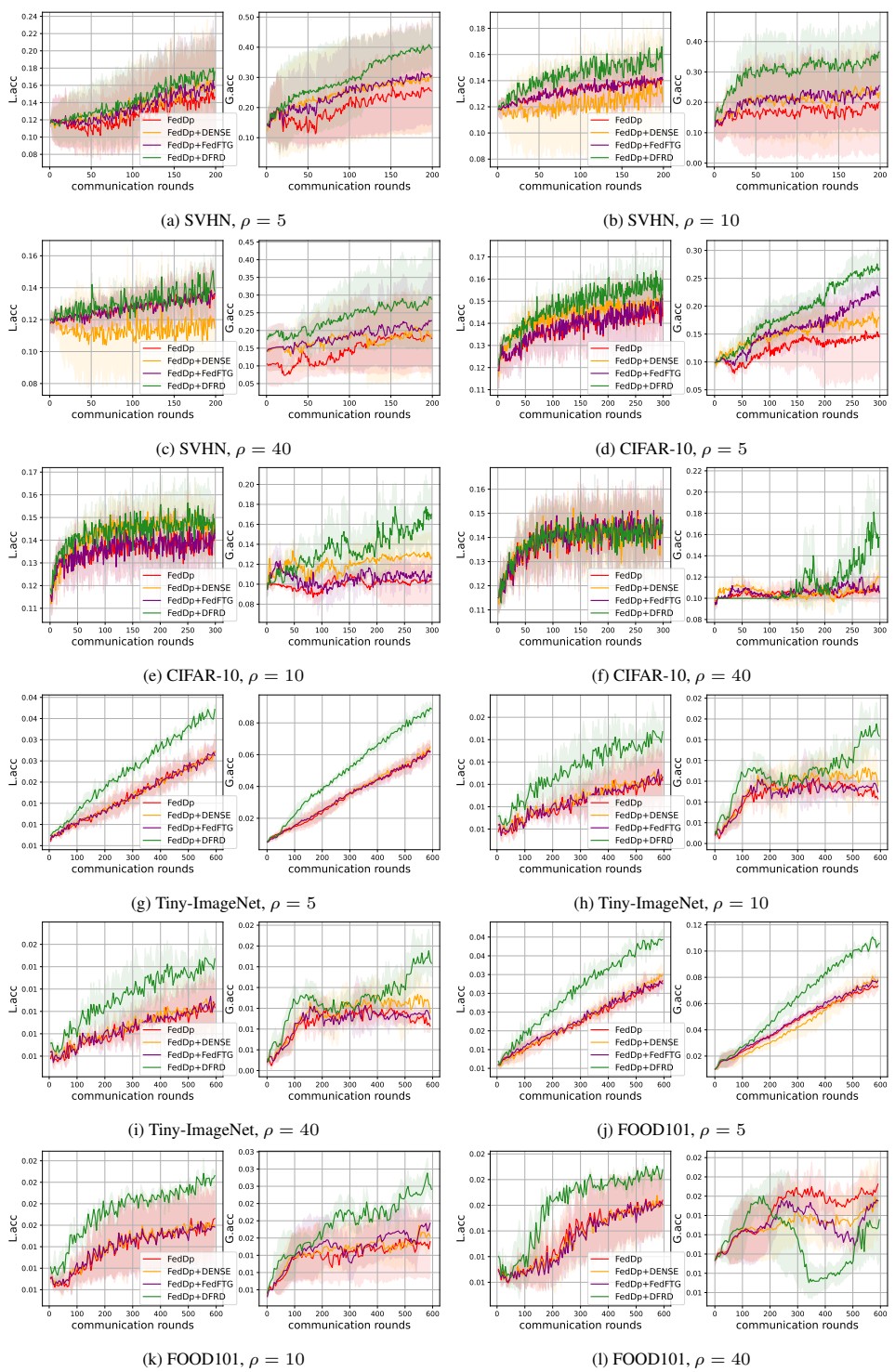

Figure 11: Learning curves of distinct fine-tuning methods based on FedDp across $\rho \in \{5, 10, 40\}$ on SVHN, CIFAR-10, Tiny-ImageNet and FOOD101.

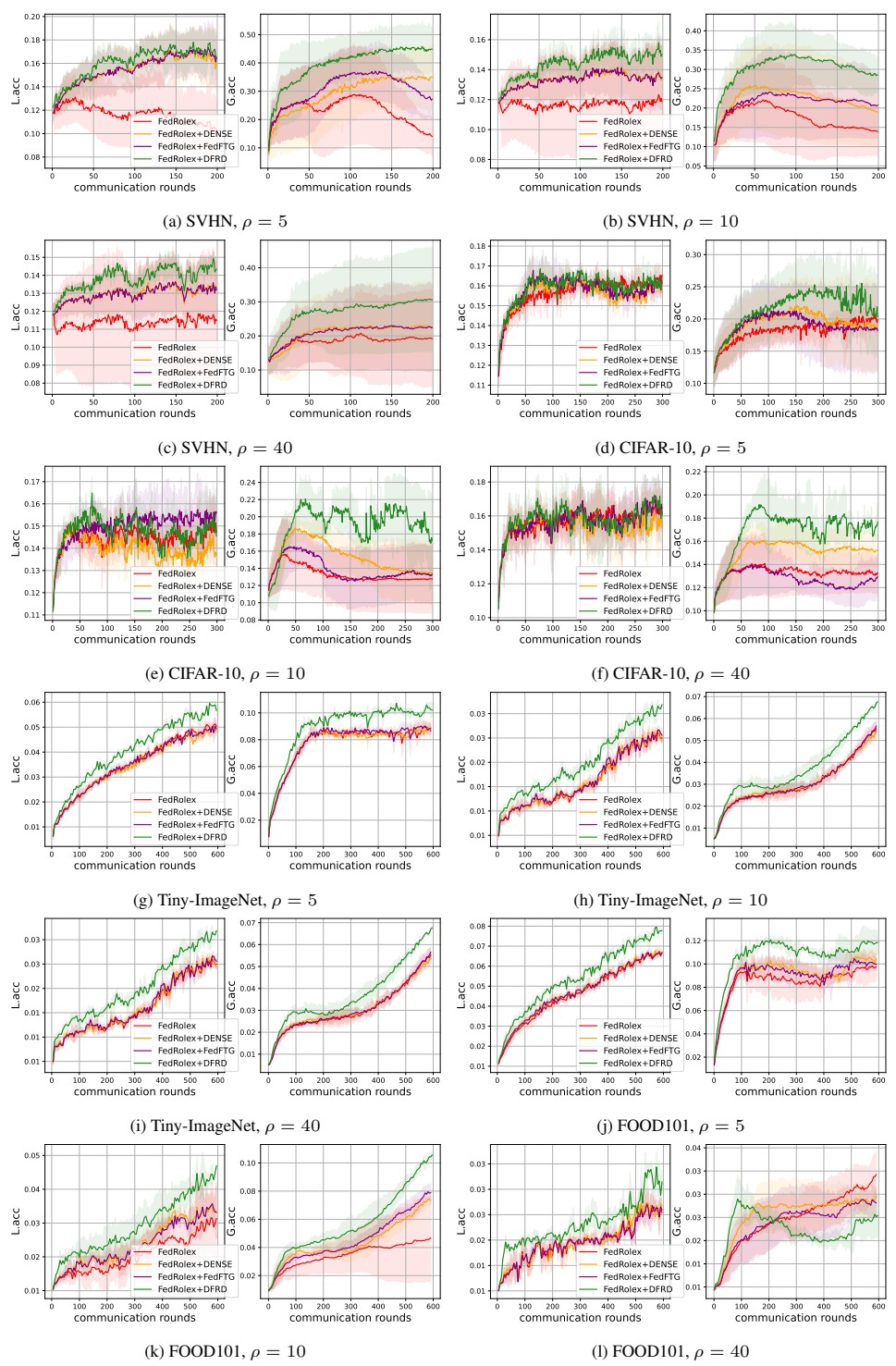

Figure 12: Learning curves of distinct fine-tuning methods based on FedRolex across $\rho \in \{5, 10, 40\}$ on SVHN, CIFAR-10, Tiny-ImageNet and FOOD101.

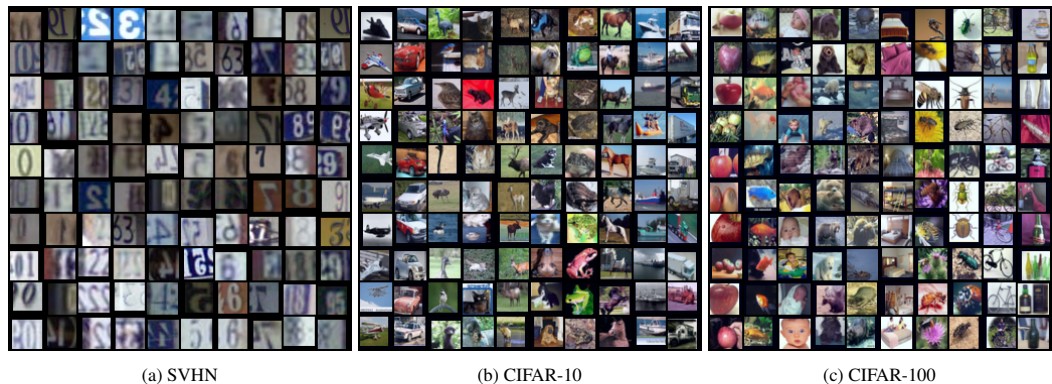

(a) SVHN        (b) CIFAR-10        (c) CIFAR-100

Figure 13: Original data selected from SVHN, CIFAR-10 and CIFAR-100. We select 10 images from each class in each dataset as a column. Note that we select the first 10 classes for display on CIFAR-100.

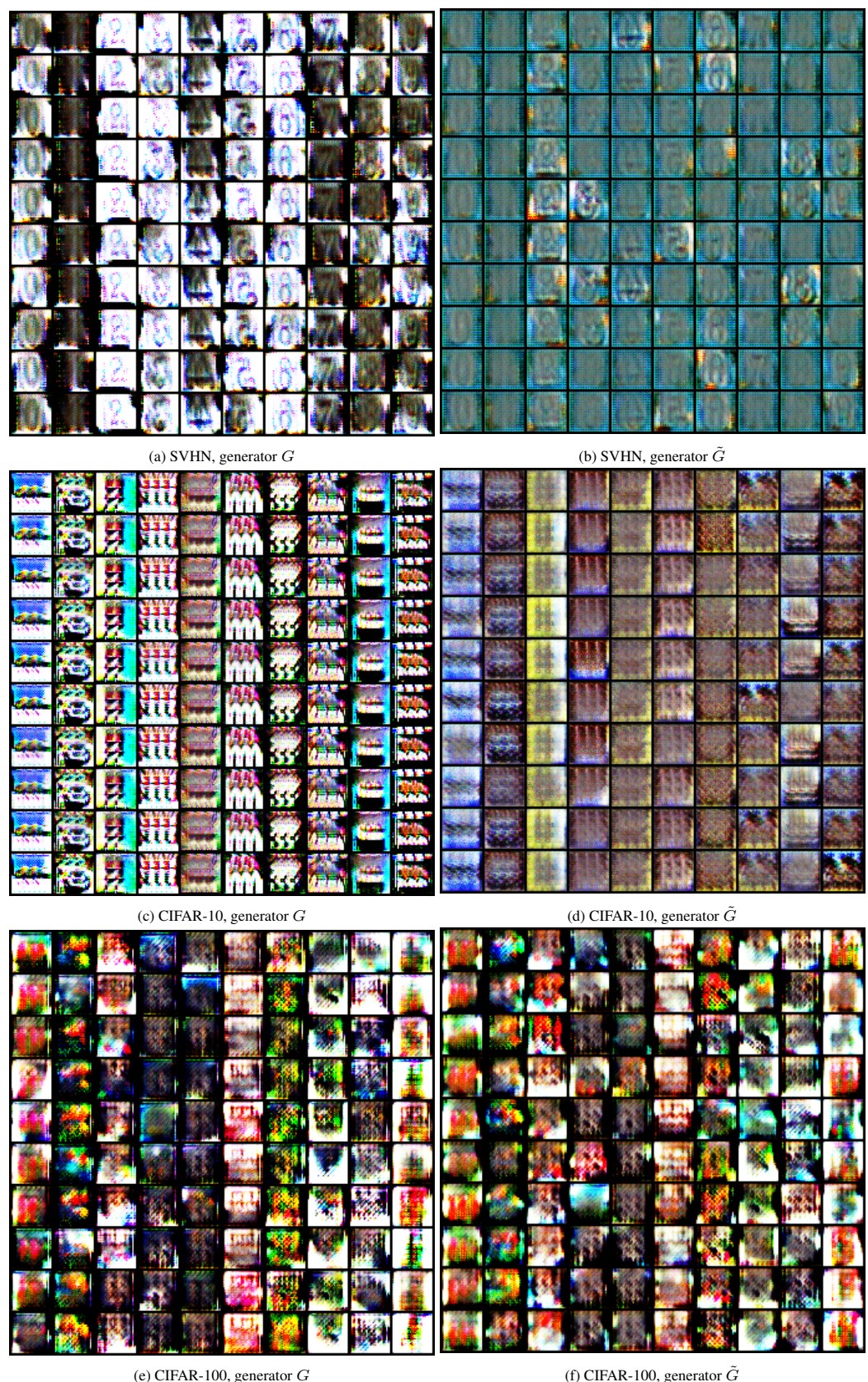

Figure 14: Visualization of synthetic data on SVHN, CIFAR-10 and CIFAR-100 by using DFRD with diversity constraint based on *mul*.

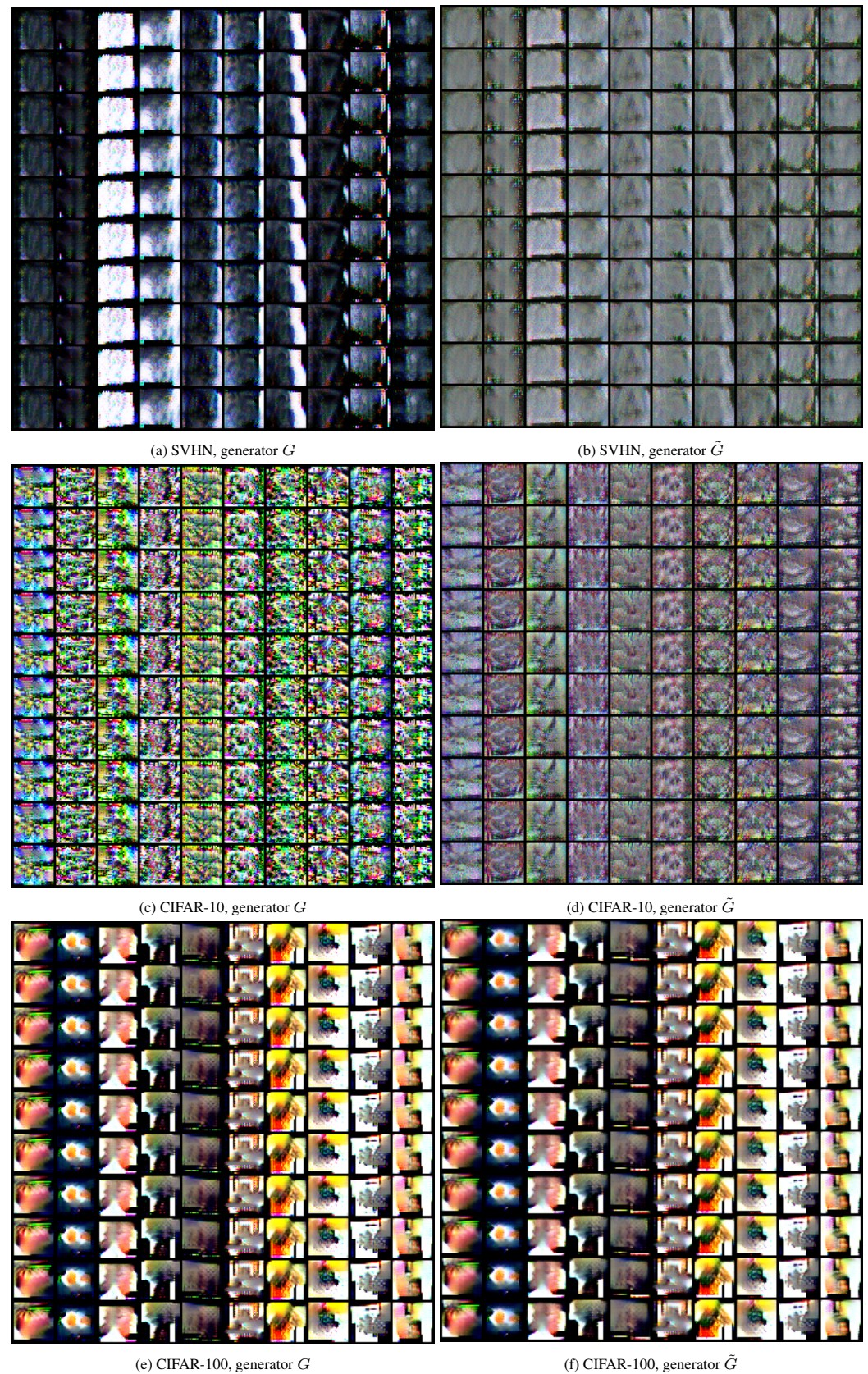

(a) SVHN, generator $G$

(b) SVHN, generator $\tilde{G}$

(c) CIFAR-10, generator $G$

(d) CIFAR-10, generator $\tilde{G}$

(e) CIFAR-100, generator $G$

(f) CIFAR-100, generator $\tilde{G}$

Figure 15: Visualization of synthetic data on SVHN, CIFAR-10 and CIFAR-100 by using DFRD with diversity constraint based on *add*.

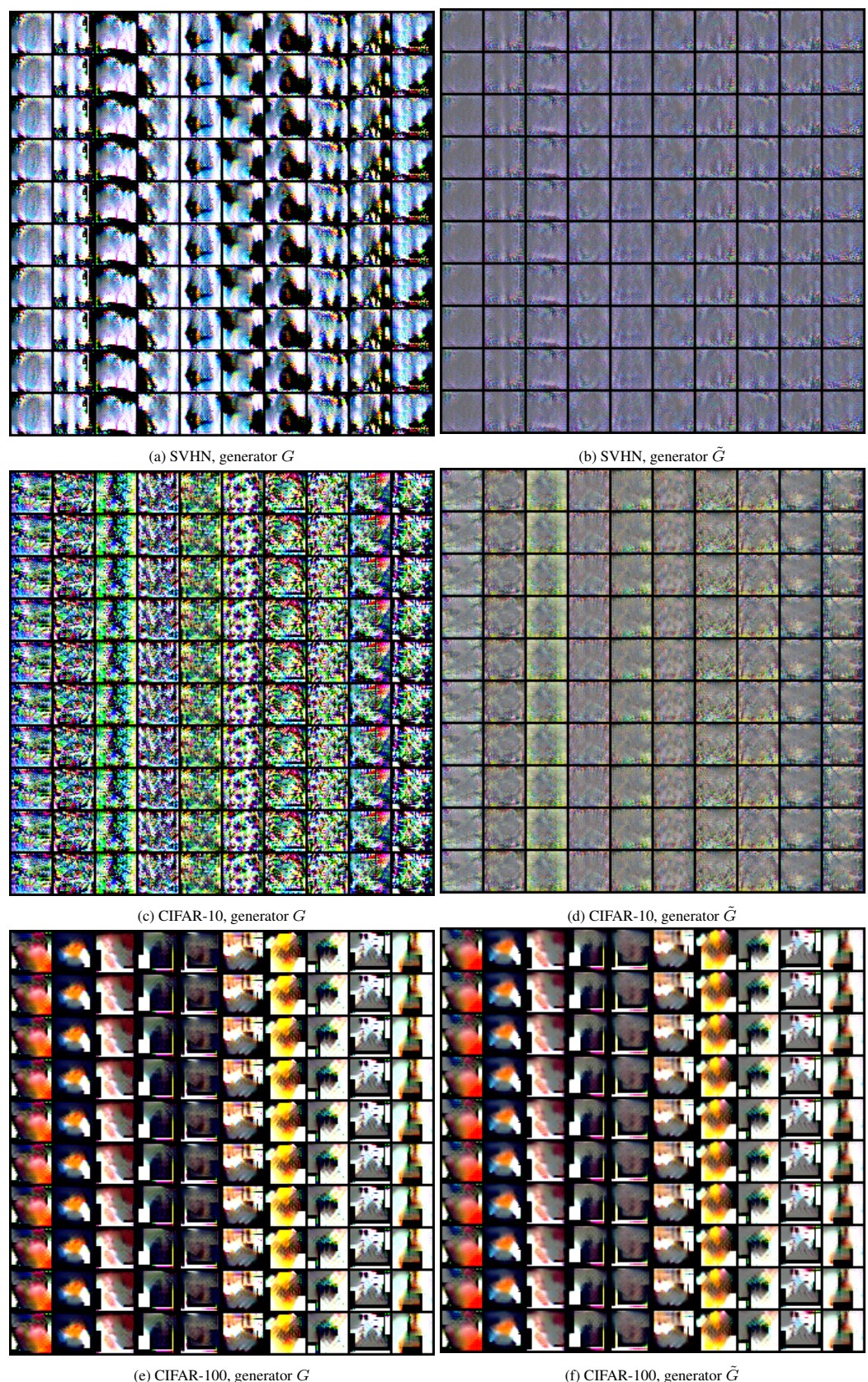

(a) SVHN, generator $G$

(b) SVHN, generator $\tilde{G}$

(c) CIFAR-10, generator $G$

(d) CIFAR-10, generator $\tilde{G}$

(e) CIFAR-100, generator $G$

(f) CIFAR-100, generator $\tilde{G}$

Figure 16: Visualization of synthetic data on SVHN, CIFAR-10 and CIFAR-100 by using DFRD with diversity constraint based on *cat*.

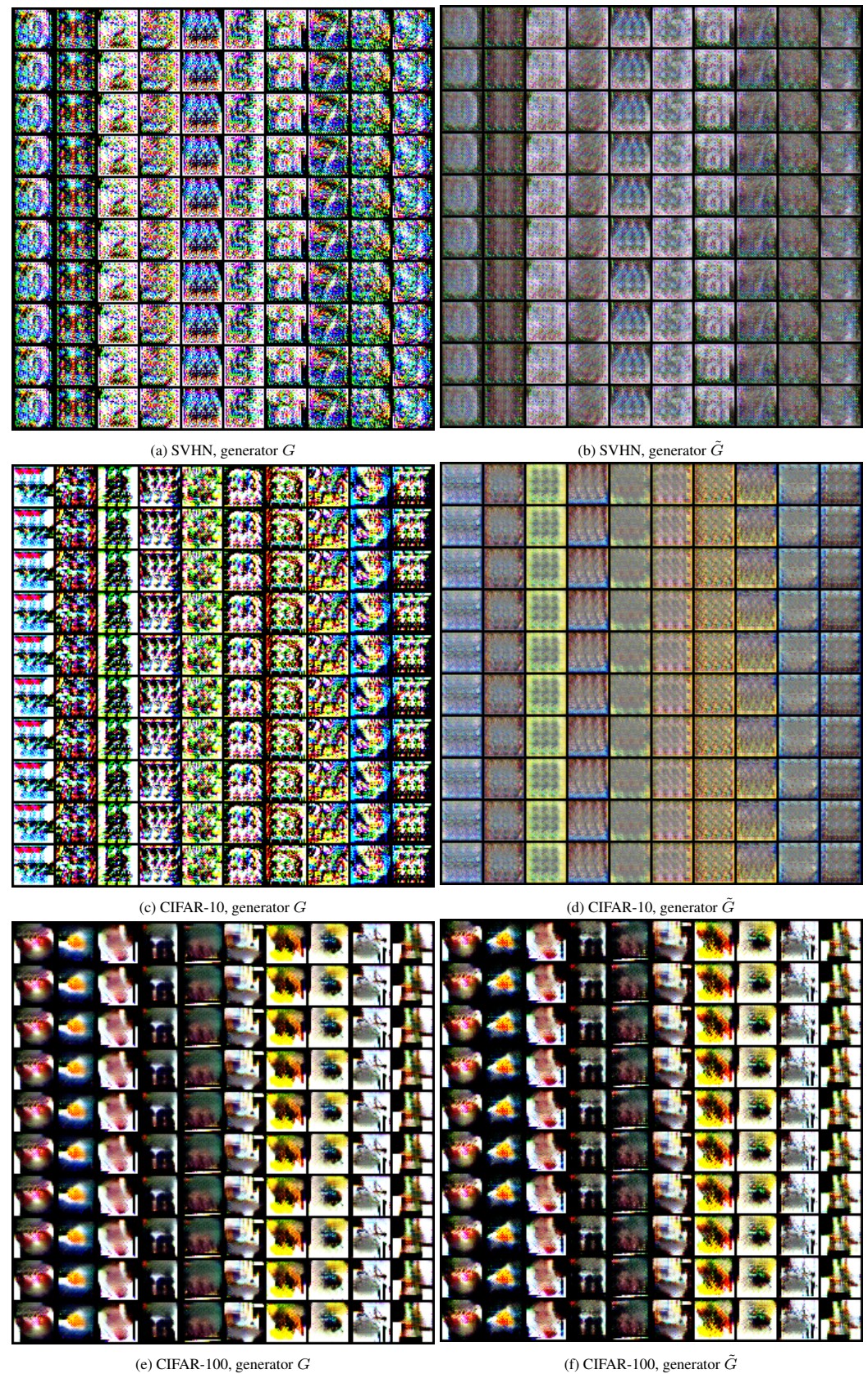

Figure 17: Visualization of synthetic data on SVHN, CIFAR-10 and CIFAR-100 by using DFRD with diversity constraint based on *n-cat*.

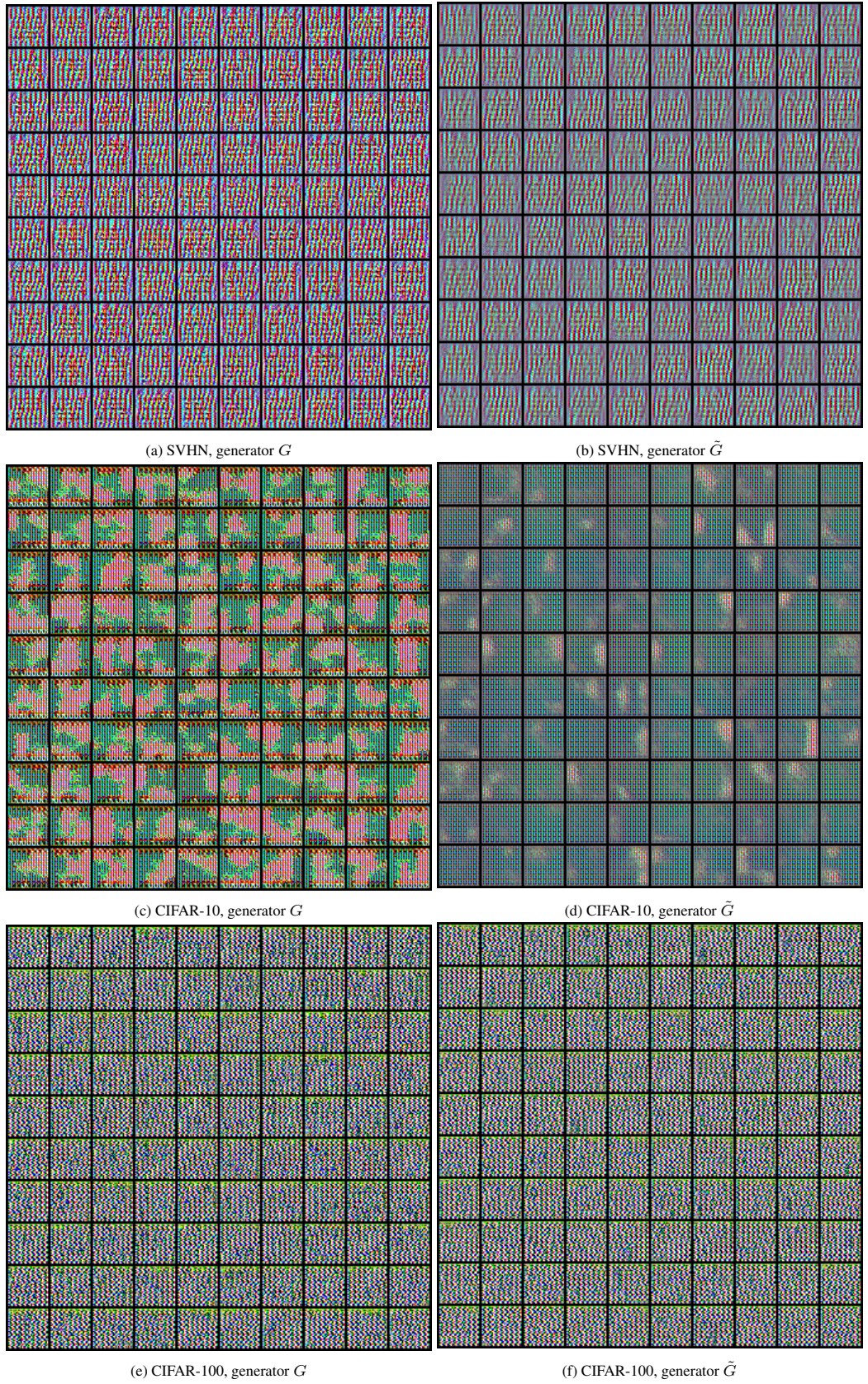

Figure 18: Visualization of synthetic data on SVHN, CIFAR-10 and CIFAR-100 by using DFRD with diversity constraint based on *none*.