# OpenReview forum: "DFRD: Data-Free Robustness Distillation for Heterogeneous Federated Learning"
_NeurIPS.cc/2023/Conference — NeurIPS 2023 poster_

### Official Review · Reviewer_deo9 · 2023-07-04

**Soundness:** 3 good
**Presentation:** 3 good
**Contribution:** 3 good
**Rating:** 6
**Confidence:** 3

**Summary:**

The overall theoretical framework of the paper can be regarded as an extension of the FedFTG theory. Building upon the server-side learning of challenging samples, it incorporates the generation of data from the previous time step by the generator to prevent catastrophic forgetting in the global model. The experimental section of the paper is substantial, and it includes comparative analyses through ablation experiments to evaluate the effectiveness of various losses. Thus, the paper makes a good contribution.

**Strengths:**

Originality: In comparison to FedFTG, the originality of the paper lies in Section 3.2, where a diverse generator is utilized by leveraging the parameter updates from the previous time step to prevent catastrophic forgetting in the model.

Quality: In comparison to FedFTG, the paper conducts thorough comparative experiments and ablative experiments that elucidate the roles of various losses, indicating a certain level of quality.

Clarity: The framework diagram, Figure 1, in the paper provides a clear explanation of the algorithm's flow.


**Weaknesses:**

3.1 The impact of knowledge transfer on synthesizing different datasets lacks theoretical support and experimental evidence, as demonstrated by the decision boundary theory presented in Figure 2. There is a lack of clear relationship between synthetic data and the decision boundaries of the student and teacher models. Since the Generator satisfies Equation 2, it is capable of generating samples (blue) on the decision boundary of the teacher model. However, it is not clearly explained under what circumstances it would generate samples beyond the decision boundaries (yellow and purple) without further elaboration.

4.3 The originality of the theoretical part of the paper is relatively weak, as using parameters from the previous time step to enhance samples appears more like a heuristic approach rather than a theoretical one.

The algorithm/objective is not concrete. For Eq(5), the objective is to minimization for which parameters?


**Questions:**

How the experiments simulate the model heterogenous?
Suggestions : The paper lacks visual experiments on samples to enhance the credibility of the theoretical training of the generator.
Parameter omega be used both for generator and Dirichlet process.

---

> ### Author Rebuttal · Authors · 2023-08-08
>
> We highly appreciate Reviewer deo9 for positive comments and precious feedback on our work. We respond to specific comments below.
>
> Q1: The impact of knowledge transfer on synthesizing different datasets lacks theoretical support and experimental evidence, as demonstrated by the decision boundary theory presented in Figure 2. There is a lack of clear relationship between synthetic data and the decision boundaries of the student and teacher models. Since the Generator satisfies Equation 2, it is capable of generating samples (blue) on the decision boundary of the teacher model. However, it is not clearly explained under what circumstances it would generate samples beyond the decision boundaries (yellow and purple) without further elaboration.
>
> R1: First, we would like to state that Figure 2 is provided as a conceptual illustration to give the reader a better understanding of our interpretations. And Figure 2 was carefully designed based on our findings and observations. In the empirical study, if the generator only satisfies Equation 2, it mainly generates synthetic data with red circles, thus making it difficult to transfer knowledge from teacher to student. Therefore, an adversarial manner is needed to generate synthetic samples with black circles, i.e., Transferability of the generator. Please see lines 128-180 and lines 211-215 for detail. Also, we clearly explain the case of generating synthetic samples with yellow and purple circles, please see lines 181-187.
>
> Q2: The algorithm/objective is not concrete. For Eq(5), the objective is to minimization for which parameters?
>
> R2: For Eq. (5), the objective is to minimize the parameters of the generator. We illustrate this in line 205 and Fig. 1.
>
> Q3: How the experiments simulate the model heterogenous? Suggestions : The paper lacks visual experiments on samples to enhance the credibility of the theoretical training of the generator. Parameter omega be used both for generator and Dirichlet process.
>
> R3: In the paper, we provide a detailed description of how to simulate heterogeneous models, see lines 257-259 and Appendix D. Additionally, we visualize synthetic samples of the generator in Figures 14-18 of Appendix.

---

> > ### Comment · Reviewer_deo9 · 2023-08-15
> > **Thank you for your response!**
> >
> > You have addressed some of my inquiries. However, the theoretical aspects concerning the boundaries of data generation still require further refinement.

---

> > > ### Author Response · Authors · 2023-08-15
> > >
> > > We greatly appreciate the feedback from the reviewers and the valuable insights provided.
> > >
> > > Thank you for emphasizing the theoretical aspects concerning the boundaries of data generation. While we concur on its significance, our paper primarily showcases practical techniques to enhance federated learning with the help DFKD.
> > > It's worth noting that even in exist well-known efforts [48, 49, 53, 56], including those on handling data and model heterogeneity for FL with the help DFKD [48, 49], comprehensive theoretical analysis concerning the boundaries of data generation is often absent.  Given the lack of suitable theoretical frameworks, we concentrated on robust empirical validation, showcasing our method (DFRD). Our results, we believe, robustly demonstrate our method's utility. We value your feedback and intend to delve deeper into theoretical aspects in future work.
> > >
> > > We aspire for our response to address the reviewer's concerns. We remain dedicated to addressing concerns you may possess with utmost eagerness.

---

> > > > ### Author Response · Authors · 2023-08-16
> > > >
> > > > Dear reviewer. I kindly inquire whether our response has successfully addressed your uncertainties. We remain dedicated to addressing concerns you may possess with utmost eagerness.

---

### Official Review · Reviewer_i49s · 2023-07-05

**Soundness:** 2 fair
**Presentation:** 2 fair
**Contribution:** 2 fair
**Rating:** 5
**Confidence:** 4

**Summary:**

The paper considers learning a robust global model in heterogeneous federated learning (FL). It aims to support scenarios of both data-heterogeneous, where data distributions among clients are non-IID (identical and independently distributed), and model-heterogeneous, where the model architecture among clients is different. To this end, the authors propose a data-free knowledge distillation (DFKD) method, which utilizes a conditional generator to generate synthetic data, which simulates the local models’ training space with respect to fidelity, transferability, and diversity. Besides, the authors utilize an exponential moving average method to mitigate the catastrophic forgetting of the global model. Experiments on real-world image classification datasets are conducted to evaluate the performance of the proposed approach.

**Strengths:**


1. The problem of heterogeneous federated learning is interesting and important.

2. A data-free knowledge distillation method with a detailed analysis of the conditional generator w.r.t. three characteristics.

3. Experiments on six real-world datasets are conducted.


**Weaknesses:**

1. The conditional generator in the proposed approach is not well motivated and elaborated.

2. The contribution of the proposed losses w.r.t. the three characteristics is incremental.

3. The choice of hyper-parameters in the exponential moving average method is unclear.

W4. Some of the experimental results are hard to comprehend.


**Questions:**

Q1. For the data heterogeneity considered in this work, what distribution the final test dataset follows? Is it the global distribution over all clients’ local data distributions?

Q2. What is the purpose of training a generator on the server for the partial training-based methods? And how is it capture the training space of various clients’ local models? More elaboration would be helpful.

Q3. Regarding the conditional generator proposed in this paper, the three characteristics (fidelity, transferability, and diversity) mainly follow those defined in [48] [49]. The contribution seems to only replace the generator with a conditional generator, which looks incremental. What is the new technical contribution in the proposed method?

Q4. The EMA method is proposed to mitigate the catastrophic forgetting of the global model. However, since the global model already captures the knowledge of local historical models, it should not forget the knowledge if it is useful. I wonder why this problem exists. Is it related to how the clients extract the local models from the aggregated global model? Besides, how to set the averaging parameters \lambda and \alpha?

Q5. It is mentioned in the paper that the generated synthetic data should be visually distinct from the real data for privacy protection. I wonder whether there is a metric to measure this. Also, if the objective is to capture the training space of local models, would it be possible to recover the clients’ local data distributions?

Q6. The proposed DFKD method is claimed to be able to address the non-IID problem; then, it would be better to compare it with solutions such as FedProx, FedNova, etc.

Q7. In Table 1 and Table 2, do the two rows of each method represent the local test accuracy and global test accuracy, or the opposite? The improvement seems to be very small compared to DENSE and FedFTG, for example, less than 1% on FMNIST.

Q8. In Figure 3a, why the global accuracy of DENSE is too low? E.g., only 10% accuracy.

Q9. In Tables 3-5, why the accuracies of the three datasets are too low? Are they well-trained or converged?


**Limitations:**

L1. It is unclear whether the proposed approach can handle the non-IID setting well, as one of the goals is to support data-heterogeneous. The authors may need to compare DFKD with non-IID solutions when the model architecture among clients is the same.

L2. The clients may not obtain useful local models due to they hold non-IID data distributions, and the server aggregates the sub-models over the global data distribution.

---

> ### Author Rebuttal · Authors · 2023-08-08
>
> We highly appreciate Reviewer i49s for positive comments and precious feedback on our work. We respond to specific comments below.
>
> Q1: For the data heterogeneity considered in this work, what distribution the final test dataset follows? Is it the global distribution over all clients’ local data distributions?
>
> R1: We detailed the setup of the test dataset in the paper, see lines 263-269.
>
> Q2: What is the purpose of training a generator on the server for the partial training-based methods? And how is it capture the training space of various clients’ local models?
>
> R2: The purpose of training a generator on the server for the partial training-based methods is to adapt the FL system to model-heterogeneous scenarios, please see lines 41-53 and 105-119. In addition, we detail how the generator captures the training space of various clients’ local models in lines 128-209.
>
> Q3: Regarding the conditional generator proposed in this paper, the three characteristics (fidelity, transferability, and diversity) mainly follow those defined in [48] [49]. The contribution seems to only replace the generator with a conditional generator, which looks incremental. What is the new technical contribution in the proposed method?
>
> R3: We acknowledge that our study of generators follows the definitions in [48, 49] in terms of fidelity, transferability, and diversity, but they do not thoroughly study the training of the generator in these three characteristics. Therefore, we propose a more effective loss objective to address their shortcomings. Please see lines 128-209 and Experiments section for detail. Moreover, our goal is to guide the generator to better capture knowledge of the local models, which cannot simply be viewed as replacing the generator with a conditional generator, since a conditional generator also is used in [48].
>
> Q4: I wonder why catastrophic forgetting problem exists. Is it related to how the clients extract the local models from the aggregated global model? Besides, how to set the averaging parameters $\lambda$ and $\alpha$?
>
> R4:  We elaborated on why catastrophic forgetting exists, see lines 211-220. Furthermore, $\lambda$ and $\alpha$ are hyperparameters that require careful fine-tuning in the experiments. We empirically study their variations on SVHN and CIFAR-10, see lines 367-383.
>
> Q5: It is mentioned in the paper that the generated synthetic data should be visually distinct from the real data for privacy protection. I wonder whether there is a metric to measure this. Also, if the objective is to capture the training space of local models, would it be possible to recover the clients’ local data distributions?
>
> R5: To our knowledge, in the current research on federated learning and data-free knowledge distillation, there are no metrics for quantifying synthetic data quality. Therefore, existing works [49, 53] use visualization of the generator's output to discriminate differences visually. Also, the generator can only partially capture the local data distribution, which is impractical to recover the clients’ local data distributions [35, 36, 53, 56]. This is because the generator does not have access to real data, it can only approximate it with the help of teacher models.
>
> Q6: The proposed DFKD method is claimed to be able to address the non-IID problem; then, it would be better to compare it with solutions such as FedProx, FedNova, etc.
>
> R6: FedProx, FedNova and so on are regularization-based methods. To ensure fair comparisons, we omit the comparison with these methods. Notably, we chose FedFTG and DENSE as baselines since they are both DFKD methods and also address the non-IID problem (please see [48] and [49] for detail).
>
> Q7: In Table 1 and Table 2, do the two rows of each method represent the local test accuracy and global test accuracy, or the opposite? The improvement seems to be very small compared to DENSE and FedFTG, for example, less than 1% on FMNIST.
>
> R7: We elaborated on what each of the two rows represents in lines 263-269. Moreover, the less than 1% improvement of our method on FMNIST compared to DENSE and FedFTG is due to the simplicity of the classification task for FMNIST. Specifically, the reason why they perform well is that FedAvg can achieve good test accuracy on FMNIST, which provides excellent initialization of the global model in each round when DENSE, FedFTG and our method are used as fine-tuning methods. However, our method significantly outperforms DENSE and FedFTG on FMNIST when they act as data-free methods rather than fine-tuning methods, see Experimental Settings and Table 1 for detail.
>
> Q8: In Figure 3a, why the global accuracy of DENSE is too low? E.g., only 10% accuracy.
>
> R8: During our empirical study, we explored the reason for the low global accuracy when DENSE is used as a data-free method rather than a fine-tuning method. We argue that DENSE is very sensitive to the initialization of the global model. Good global model initialization can greatly improve the performance of DENSE. For example, FedAvg+DENSE can achieve good accuracies, see Table 1.
>
> Q9: In Tables 3-5, why the accuracies of the three datasets are too low? Are they well-trained or converged?
>
> R9: The low accuracy rates in Tables 3-5 are related to our experimental setup. We perform the ablation study in both data and model heterogeneous settings, please see lines 299-303 and the Experimental Setting section for detail. Also, they’re well-trained.
>
> Q10: It is unclear whether the proposed approach can handle the non-IID setting well, as one of the goals is to support data-heterogeneous. The authors may need to compare DFKD with non-IID solutions when the model architecture among clients is the same.
>
> R10: Our method proposes Dynamic Weighting and Label Sampling to handle the non-IID setup, see lines 231-236 and Appendix F for detail.

---

> > ### Author Response · Authors · 2023-08-16
> >
> > Dear reviewer. I kindly inquire whether our response has successfully addressed your uncertainties. We remain dedicated to addressing concerns you may possess with utmost eagerness.

---

> > > ### Comment · Reviewer_i49s · 2023-08-16
> > >
> > > Thanks for the response, which has clarified some of my concerns.
> > >
> > > For R1, the authors only describe the setting used in the experiments, which I am unsure whether it covers the setting supported by the proposed as the experimental setting could be a special case. It is better to clarify the scenario considered in this paper together with the problem formulation description and justify its practicality. For example, in which applications the clients may share the same test data distribution while having different training data distributions? Or can the proposed method also handle the setting where different clients have their own test data? With these, the problem shall be better motivated.
> > >
> > > For R2, lines 41-53 and lines 105-119 only briefly describe the partial-training methods; they are not related to the generator training. Also, lines 128-209 present the characteristics of a well-behaved generator; however, it is unclear to me why training such a generator and combining it with the PT-based method helps improve its effectiveness. In other words, the connection between the four stages in DFRD could be further improved, e.g., by providing more intuitive explanations.
> > >
> > > For R6, it is not convincing for  the authors to omit the comparison because the others are regularization-based solutions. Since they are also solutions to address the non-iid problem in FL, it is straightforward to compare with them, though they do not consider the hardware resource heterogeneity. Moreover, I found that FedFTG [48] also compared with FedProx, SCAFFOLD, etc., and FedFTG outperforms those solutions on CIFAR-10 and CIFAR-100 datasets with respect to several data heterogeneity settings. I am not sure if the experimental setting is the same as that in the FedFTG paper, but it seems the accuracy of FedFTG in [48] is much higher than the ones reported in this paper. For example, under the iid setting on CIFAR-10, the accuracy of FedFTG in [48] is around 87%, while that in this paper is only around 47%. Is it because in the experiments of this paper, the authors also simulate the model heterogeneity, which affects the accuracy? If so, what would be the result if the proposed DFRD method follows the same setting as that in FedFTG? If it also outperforms FedFTG, then it should not be necessary to compare to those regularization-based solutions, as they are already demonstrated to be inferior to FedFTG.
> > >
> > > For R9, would it be possible to further explain what experimental setup cause low accuracy and the possible reasons? Otherwise, the results look ad-hoc and are difficult to predict.
> > >
> > > For R10, it would be better to briefly elaborate on how Dynamic Weighting and Label Sampling handle the non-IID setup in the main manuscript, as this part is important.
> > >
> > > I therefore keep my score.

---

> > > > ### Author Response · Authors · 2023-08-16
> > > >
> > > > We greatly appreciate the feedback from the reviewers and the valuable insights provided. We respond to specific comments below.
> > > >
> > > > R1:
> > > > Our experimental setup is applicable to many real-world scenarios.
> > > > For example, the application scenarios we focus on include the following:
> > > >
> > > > Mobile Devices[1]: Federated Learning can be applied on mobile devices where users may encounter similar tasks such as speech recognition or image classification. However, their training data may vary due to differences in personal preferences or geographic locations.
> > > >
> > > >
> > > > Internet of Things (IoT)[2]: in the IoT domain, different devices may share the same test data distribution, such as sensor data or environmental monitoring data. However, their training data may differ in data distribution due to factors such as device installation locations, environmental conditions, etc., such as meteorological data or environmental parameters in different regions.
> > > >
> > > > Autonomous Driving[3]: Within autonomous driving systems, various vehicles may share the same distribution of road and traffic scene test data to conduct driving decision tests in different environments. However, the training data for each vehicle, including sensor data and vehicle performance parameters, may differ due to factors such as vehicle model and manufacturer.
> > > >
> > > > Our method can be applied to the setting where different clients have their own test data, but we acknowledge that our method may not be able to handle this setting.
> > > > Furthermore, we deem that scenarios where different clients possess their own test data fall under the domain of personalized federated learning, which is not the focus of our work. This is because the goal of DFRD trains a robust global model.
> > > >
> > > >
> > > > [1] W. Y. B., Luong, N. C., Hoang, D. T., Jiao, Y., Liang, Y. C., Yang, Q., Niyato, D.,  & Miao, C. (2020) Federated learning in mobile edge networks: A comprehensive survey.  IEEE Communications Surveys & Tutorials, 22(3): 2031-2063.
> > > >
> > > > [2] Nguyen, D. C., Ding, M., Pathirana, P. N., Seneviratne, A., Li, J., & Poor, H. V.  (2021) Federated learning for internet of things: A comprehensive survey. IEEE Communications Surveys & Tutorials, 23(3): 1622-1658.
> > > >
> > > > [3] Li, Y., Tao, X., Zhang, X., Liu, J., & Xu, J.  (2021) Privacy-preserved federated learning for autonomous driving. IEEE Transactions on Intelligent Transportation Systems, 23(7): 8423-8434.
> > > >
> > > > R2:
> > > > From lines 41-53 and 105-119, it is evident that the PT-based methods work on model-heterogeneous scenarios.
> > > > Specifically, PT-based methods send width-based sub-models to clients, which are extracted by the server from a larger global model according to each client's resource budget, and then aggregate these trained sub-models to update the global model.
> > > > PT-based methods can be considered as an extension of FedAvg to model-heterogeneous scenarios, which means they are implementation-friendly and computationally efficient, but they also suffer from the same adverse effects from data heterogeneity as FedAvg or even more severely.
> > > >
> > > > Meanwhile, existing studies FedFTG and DENSE have verified that data heterogeneity can be effectively mitigated with the help of data-free knowledge distillation (DFKD), thereby enhancing the performance of the global model obtained through FedAvg.
> > > > Therefore, we aim to improve the performance of global models obtained by PT-based methods by leveraging DFKD.
> > > >
> > > > In particular, the generator serves as a crucial component of DFKD, emulating the input space of the local models guided by the objective loss we formulate (as shown in Equation 5), and providing data for the subsequent model distillation (as shown in Equation 7).
> > > > This is why we need to learn a well-trained generator and combine it with PT-based methods to enhance the performance of the global model.
> > > >
> > > > Moreover, we appreciate the reviewer's suggestion, and we will provide more intuitive explanations in the revised main paper to facilitate better understanding.
> > > >
> > > > Due to space constraints, please proceed to the next response for further information. Thank you very much.

---

> > > > > ### Author Response · Authors · 2023-08-16
> > > > >
> > > > > R6:
> > > > > For clarification, we illustrate the experimental setup for Tables 1 and 2. In Table 1, we investigate the impact of different levels of data heterogeneity (i.e., $\omega \in \{0.01, 0.1, 1.0\}$) under the model homogeneous scenario. In Table 2, under the data heterogeneity level of $\omega=0.1$, we examine the influence of varying levels of model heterogeneity (i.e., $\rho \in \{5, 10, 40\}$). Due to space limitations in the main paper, we have provided detailed description in lines 686 to 707 of the Appendix.
> > > > > In addition, FedFTG leverages DFKD to fine-tune the global model in model-homogeneous FL to overcome data heterogeneity. This corresponds to FedAvg+FedFTG as described in our paper. To avoid ambiguity, we have provided explanations in lines 246 to 254.
> > > > > From the above, it can be seen that the results in Table 1 are obtained under a model homogeneous scenario, which is consistent with [48].
> > > > >
> > > > > For CIFAR-10, we differ from [48] in experimental settings, such as model architecture, number of clients, and selection of data heterogeneity levels. Please refer to lines 688 to 693 for specific details.
> > > > > This results in differences between our experimental results and those of [48].
> > > > > For example, the accuracy of FedAvg+FedFTG in Table 1 is 68.85% (i.e., $\omega=1.0$) in Table 1.
> > > > > However, this does not affect the validation that our method (FedAvg+DFRD) outperforms FedAvg+FedFTG in model-homogeneous scenario.
> > > > > It is worth noting that for CIFAR-100, considering the differences in experimental settings such as the number of clients and data heterogeneity levels, the accuracy of FedAvg+FedFTG in Table 1 is 58.81% (i.e., $\omega=1.0$), while the accuracy reported in [48] is 56.94% (i.e., iid).
> > > > >
> > > > > As for why we did not consider the regularization-based methods as our baselines, it is because we are focused on enhancing the performance of the global model through DFKD, and we have demonstrated that FedAvg+DFRD outperforms FedAvg+FedFTG in model-homogeneous scenario.
> > > > > This is also the reason we mention to ensure a fair comparison.
> > > > >
> > > > >
> > > > > R9：The low accuracy in Tables 3-5 is due to our experimental setup, where we consider both data and model heterogeneity.
> > > > > For example, in the case of CIFAR-10, we consider data heterogeneity with $\omega=0.1$ (see Figure 6 in the Appendix for visualized data heterogeneity distribution) and model heterogeneity with $\rho=10$ (model heterogeneity distribution is provided in line 649 in the Appendix). Subsequently, we trained adequately for the ablation study.
> > > > > In the challenging experimental settings mentioned above, the global model not only needs to handle client drift caused by data heterogeneity but also suffers from the weak learning capabilities of low-end models (e.g., 1/16-width capacity w.r.t. the global model).
> > > > > This is the reason behind the low accuracy in Tables 3-5.
> > > > >
> > > > >
> > > > > R10:
> > > > > We appreciate the reviewer's suggestion, and we will enhance the explanation and clarification of Dynamic Weighting and Label Sampling in revised main paper.
> > > > >
> > > > > We aspire for our response to address the reviewer's concerns. We remain dedicated to addressing concerns you may possess with utmost eagerness.

---

### Official Review · Reviewer_XBHz · 2023-07-05

**Soundness:** 3 good
**Presentation:** 2 fair
**Contribution:** 3 good
**Rating:** 5
**Confidence:** 4

**Summary:**

This paper studies how to learn a robust global model in the data-heterogeneous and model-heterogeneous FL, by proposing a data-free knowledge distillation method. The proposed method is evaluated on six image classification datasets and outperforms compared methods.

**Strengths:**

- This paper studies data heterogeneity and model heterogeneity in FL, which are relevant and important topics.
- The motivation for solving both data and model heterogeneity are well explained.
- The proposed method outperforms compared methods.
- The authors conduct many ablation studies.


**Weaknesses:**

- It is not clear how to calculate the term \tau_{i,y}, which is important to adjust the synthetic data.
- Only intuitive toy visualizations are shown. It lacks experimental visualizations to support the mentioned “Fidelity” of the proposed method.
- The experiments only validate the model heterogeneity in terms of width.
- The results presentation can be improved. The table is small, and some symbols are confusing.
- The experiments are mainly conducted on datasets with small image sizes （e.g., 32*32 or 64*64), lack of validation on large datasets.
- The global accuracy is sensitive to different values of \beta, and the standard deviation is large.


**Questions:**

- Why do DFRD and +DFRD show different trends in Table 1? For DFRD, L.acc is higher than G.acc, but for +DFRD, L.acc is lower than G.acc.
- Is this method applicable to other model heterogeneity cases? Such as different network architecture?
- Why replace FMNIST in table.1 with TinyImagnet in table.2?


**Limitations:**

The authors discussed the limitations with potential solutions, and the broader impacts.

---

> ### Author Rebuttal · Authors · 2023-08-08
>
> We highly appreciate Reviewer XBHz for positive comments and precious feedback on our work. We respond to specific comments below.
>
> Q1: It is not clear how to calculate the term $\tau_{i,y}$, which is important to adjust the synthetic data.
>
> R1: We delve into the calculation of $\tau_{i,y}$ in the paper, see lines 231-236 and Appendix F.
>
> Q2: Only intuitive toy visualizations are shown. It lacks experimental visualizations to support the mentioned “Fidelity” of the proposed method.
>
> R2: We visualize the synthetic data of generators with different diversity constraints over SVHN, CIFAR-10, and CIFAR-100 in Figures 14-18 in the Appendix, where Figure 14 is the visualization results of our proposed method. From Figure 14, we can observe that the synthetic data generated by the generators exhibits significant differences among the distinct classes, demonstrating the high fidelity of the proposed method.
>
> Q3: The experiments are mainly conducted on datasets with small image sizes （e.g., 3232 or 6464), lack of validation on large datasets.
>
> R3: We agree with this point. However, all our experiments are conducted on one NVIDIA Tesla A100 GPU with 80Gb of memory. We have tried datasets with image sizes of 128x128 and 256x256 in our experimental device, but running DFRD and baselines on these datasets would incur extreme time costs. So, we choose datasets with small image sizes for our experimental study.
>
> Q4: Why do DFRD and +DFRD show different trends in Table 1? For DFRD, L.acc is higher than G.acc, but for +DFRD, L.acc is lower than G.acc.
>
> R4: We explain the above phenomena in the paper, please see lines 271-281.
>
> Q5: Is this method applicable to other model heterogeneity cases? Such as different network architecture?
>
> R5: Our method can be applied to other cases of model heterogeneity, such as different network architectures.
>
> Q6: Why replace FMNIST in table.1 with TinyImagnet in table.2?
>
> R6: On the one hand, we want to study the performances of our method and baselines on datasets with more difficult classification tasks. On the other hand, Tables 1 and 2 are about data heterogeneity and model heterogeneity, respectively, so there is no need to force a fixed dataset. Another important reason is the space limitation of the main paper.
>
> Other suggestions provided by the reviewer to help improve the paper will be corrected in the revised paper.

---

> > ### Comment · Reviewer_XBHz · 2023-08-14
> >
> > Thanks for the authors' response; some of my concerns have been addressed. However, the weaknesses of point 3 and point 6 have not been responded. I would like to keep my ratings unchanged.

---

> > > ### Author Response · Authors · 2023-08-15
> > >
> > > We greatly appreciate the feedback from the reviewers and the valuable insights provided.
> > >
> > > We thank the reviewer for raising the point 3 and point 6 on the weakness section. We noted them during the rebuttal. To address the reviewer's concerns, we provide a careful response for the point 3 and 6.
> > >
> > > For point 3, we acknowledge that we only validated model heterogeneity in terms of width. However, we do mention in our paper that our primary focus lies in addressing model heterogeneity through the PT-based method [28-31], which represents a significant category of methods to tackle model heterogeneity. Please refer to lines 42, 47-52, and 105-119 for more details. Indeed, there exist prior works [60] and [61] that study model heterogeneity in terms of depth, but inclusion of intricate heuristic operations restricts their applicability. Hence, we lean towards PT-based method. Furthermore, DENSE studies model heterogeneity scenarios wherein the architectures of local models are entirely dissimilar, and can only transfer knowledge from the local models to the randomly initialized global model, resulting in a degradation of the global model's performance, as illustrated in Table 1.
> > >
> > > For point 6, we thank for pointing out the details of Fig 4. From Fig. 4, we can see that DFRD maintains stable test performance among all selections of $\beta_{tran}$ and $\beta_{div}$ over SVHN.  Meanwhile, $G.acc$ fluctuates slightly with the increases of $\beta_{tran}$ and $\beta_{div}$ on CIFAR-10. The above results indicate that DFRD is not sensitive to choices of $\beta_{tran}$ and $\beta_{div}$ over a wide range.
> > > Also, the large standard deviation is not due to $\beta_{tran}$ and $\beta_{div}$, but stems from our experimental setup. Specifically, to ensure reliability, we report the average for each experiment over $3$ different random seeds. Different seeds imply different heterogeneity (including data and model) and model initialization, so our experimental setups vary greatly among seeds, resulting in large standard deviations.  Moreover, in our extensive comparison experiments, DFRD achieves the smallest standard deviation compared to baselines in most cases, which robustly demonstrates our method's utility.
> > >
> > > We aspire for our response to address the reviewer's concerns. We remain dedicated to addressing concerns you may possess with utmost eagerness.

---

> > > > ### Author Response · Authors · 2023-08-16
> > > >
> > > > Dear reviewer. I kindly inquire whether our response has successfully addressed your uncertainties. We remain dedicated to addressing concerns you may possess with utmost eagerness.

---

> > > > > ### Comment · Reviewer_XBHz · 2023-08-19
> > > > >
> > > > > Many thanks for the authors' detailed explanations. I have no more questions.

---

> > > > > > ### Author Response · Authors · 2023-08-19
> > > > > >
> > > > > > We are pleased to be able to address the reviewer's concerns. We remain dedicated to addressing concerns you may possess with utmost eagerness. Additionally, we would be encouraged if the reviewer considers raising the score.

---

### Official Review · Reviewer_1PRp · 2023-07-06

**Soundness:** 4 excellent
**Presentation:** 4 excellent
**Contribution:** 3 good
**Rating:** 6
**Confidence:** 4

**Summary:**

In the presented paper, the authors lay out a strategy for fine-tuning a global model, specifically within the Heterogeneous Federated Learning setting. To transfer knowledge from the individual local models (identified as 'teachers') to the global model (or 'student'), the authors construct and train a generative model that is capable of producing pseudo samples, inspired from the Data-Free Knowledge Distillation literature.

Furthermore, the authors suggest preserving an exponentially moving copy of this generative model. This is specifically designed to provide pseudo-samples from prior communication rounds. The overarching goal of this method is to protect the global model from catastrophic forgetting - a substantial problem in machine learning where a model, after learning new information, forgets the old. By accessing these pseudo-samples from previous rounds, the global model can effectively reinforce its learning and maintain a more comprehensive understanding over the communication rounds.

Lastly, the authors introduce a system that employs dynamic weighting and label sampling. This is proposed to enhance the precision in extracting knowledge from local models. By implementing this technique, it is intended that the knowledge transfer process will be more accurate and effective.


**Strengths:**

1. In this manuscript, the authors introduce an elegant yet straightforward technique to avert catastrophic forgetting, an issue that often arises in machine learning models, by maintaining a EMA generator. Remarkably, the method appears to maintain a consistent spatial footprint, equivalent to the dimensions of the generator models, throughout the various communication cycles. This ingenious mechanism facilitates the generation of pseudo samples from preceding rounds, thereby extending its usefulness and practicality in real-world applications.

2. Moreover, the depth of empirical examination carried out in this paper is commendable. The authors have meticulously analyzed the performance of their technique, employing a thorough sensitivity analysis and ablation studies to evaluate the impact of the individual components of their methodology. This level of rigor and detail, often overlooked, attests to the quality of the research conducted and significantly boosts the credibility of their findings.


**Weaknesses:**

1. The manuscript offers a promising exploration into mitigating catastrophic forgetting in the context of Federated Heterogeneous Learning. However, as a reviewer, I find that the examination of catastrophic forgetting and its severity within this specific context could have been more in-depth. Works such as those by Binci et al. [1,2] and Patel et al. [3], which tackle catastrophic forgetting in adversarial settings, provide comprehensive empirical evidence via monitoring student learning curves. Such an approach would have deepened the current study's investigation, allowing readers to better appreciate its significance.

2. The current research draws substantially from existing methodologies, including those used in FedFTG [4] and DENSE [5], particularly adopting the data-free knowledge transfer framework from Do et al. [6]. Consequently, it becomes challenging to discern the novel contribution of this paper. Greater emphasis on aspects like Dynamic Weighting and Label Sampling, currently found in the appendix, would have added more substance to the primary narrative of the manuscript. Additionally, a thorough exploration of the obstacles faced when adapting the adversarial data-free knowledge transfer framework [6] to a Heterogeneous FL setting would have enriched the study.

Minor:
1. In the related works section, the absence of adversarial DFKD methodologies, such as those in [1,2,3], which also aim to prevent catastrophic forgetting, is a notable omission. I suggest that the authors provide a clear rationale for their preference for the DFKD framework in [6] over the others ([3] could be excluded due its recentness), clarifying this would enhance the comprehensibility of the manuscript.


References:

[1] Kuluhan Binci, Nam Trung Pham, Tulika Mitra, and Karianto Leman. "Preventing catastrophic forgetting and distribution mismatch in knowledge distillation via synthetic data." In WACV, 2022.

[2] Kuluhan Binci, Shivam Aggarwal, Nam Trung Pham, Karianto Leman, and Tulika Mitra. "Robust and resource-efficient data-free knowledge distillation by generative pseudo replay." In AAAI, 2022.

[3] Gaurav Patel, Konda Reddy Mopuri, and Qiang Qiu. "Learning to Retain while Acquiring: Combating Distribution-Shift in Adversarial Data-Free Knowledge Distillation." In CVPR, 2023.

[4] Lin Zhang, Li Shen, Liang Ding, Dacheng Tao, and Ling-Yu Duan. "Fine-tuning global model via data-free knowledge distillation for non-iid federated learning." In CVPR, 2022.

[5] Jie Zhang, Chen Chen, Bo Li, Lingjuan Lyu, Shuang Wu, Shouhong Ding, Chunhua Shen, and Chao Wu. "Dense: Data-free one-shot federated learning." In NeurIPS, 2022.

[6] Kien Do, Thai Hung Le, Dung Nguyen, Dang Nguyen, Haripriya Harikumar, Truyen Tran, Santu Rana, and Svetha Venkatesh. "Momentum Adversarial Distillation: Handling Large Distribution Shifts in Data-Free Knowledge Distillation." In NeurIPS, 2022.


**Questions:**

1. I am interested in gaining a deeper understanding of the process behind the observations and interpretations, specifically those pertaining to transferability, decision boundaries, and the positioning of synthetic data-points (L165-180). Could the authors elaborate on the methodology or rationale that led to these interpretations?

2. Furthermore, the visualization provided in Figure 2 has caught my attention. I would appreciate it if the authors could elucidate how this figure was constructed. Does this visual representation arise from actual data obtained through a small-scale experiment, or is it a conceptual illustration reflecting your interpretations?

3. Regarding equation 5, I noticed that the KL divergence loss is applied directly to the logits (pre-softmax) features, as opposed to the prediction distribution resulting from the application of softmax to the logits. Could the authors provide a comprehensive explanation for this choice? Does this approach hold any specific advantages or implications for the overall results of your study?


**Limitations:**

The authors have thoroughly listed out the limitations and broader impacts in the appendix.

---

> ### Author Rebuttal · Authors · 2023-08-08
>
> We highly appreciate Reviewer 1PRp for positive comments and precious feedback on our work. We respond to specific comments below.
>
> Q1: The manuscript offers a promising exploration into mitigating catastrophic forgetting in the context of Federated Heterogeneous Learning. However, as a reviewer, I find that the examination of catastrophic forgetting and its severity within this specific context could have been more in-depth. Works such as those by Binci et al. [1,2] and Patel et al. [3], which tackle catastrophic forgetting in adversarial settings, provide comprehensive empirical evidence via monitoring student learning curves. Such an approach would have deepened the current study's investigation, allowing readers to better appreciate its significance.
>
> R1: We thank the reviewer's constructive suggestion. We will refer to [1], [2] and [3] as references, thus deepening the investigation that addresses the problem of catastrophic forgetting in adversarial settings. Also, compare to [1], [2] and [3], [6] is more memory efficient, adapts better to the student update, and is more stable for a continuous stream of synthetic samples. That's why we prefer [6]. We will update them in the revised paper, thereby enhancing the comprehensibility of the manuscript.
>
> Q2: I am interested in gaining a deeper understanding of the process behind the observations and interpretations, specifically those pertaining to transferability, decision boundaries, and the positioning of synthetic data-points (L165-180). Could the authors elaborate on the methodology or rationale that led to these interpretations?
>
> R2: As we proceeded with our work, we first delved into existing related works on transferability including references [4], [5], and [6], among others. Then, during the empirical study, we found that the generator produces two types of synthetic data with yellow and purple circles in Fig. 2 (c), which can mislead the generator. For example, the conditional generator $G(\cdot)$ takes label information (e.g., $y=1$) as one of the inputs to generate synthetic data s. However, the ensemble model’s inferred output based on s results in label 2, i.e., $y=2$. This inconsistency degrades the quality of synthetic data generated by $G(\cdot)$, thereby negatively affecting the global model's performance, as shown in Table 3. The aforementioned process and principles elucidate how we led to these interpretations.
>
> Q3: Furthermore, the visualization provided in Figure 2 has caught my attention. I would appreciate it if the authors could elucidate how this figure was constructed. Does this visual representation arise from actual data obtained through a small-scale experiment, or is it a conceptual illustration reflecting your interpretations?
>
> R3: Figure 2 is a conceptual illustration to give the reader a better understanding of our interpretations. And Figure 2 was carefully designed based on our findings and observations during our empirical study.
>
> Q4: Regarding equation 5, I noticed that the KL divergence loss is applied directly to the logits (pre-softmax) features, as opposed to the prediction distribution resulting from the application of softmax to the logits. Could the authors provide a comprehensive explanation for this choice? Does this approach hold any specific advantages or implications for the overall results of your study?
>
> R4: In our work, KL divergence loss is applied to the prediction distribution resulting from the application of softmax to the logits, as opposed to the logits (pre-softmax) features. Both softmax and log-softmax operations are considered within the KL divergence (please see our code). We will clear them further in the revised paper for ease of understanding.

---

> > ### Comment · Reviewer_1PRp · 2023-08-13
> > **Thanks for the response!**
> >
> > Thank you to the authors for their comprehensive response to the review(s). They have addressed most of my concerns. I would like to maintain my initial rating as I stand by the second point on the weakness section. Nonetheless, I acknowledge the non-triviality of the problems faced and tackled by the authors.

---

> > > ### Author Response · Authors · 2023-08-15
> > >
> > > We greatly appreciate the feedback from the reviewers and the valuable insights provided.
> > >
> > > We thank the reviewer for raising the second point on the weakness section. We noted it during the rebuttal and have made adjustments and additions accordingly.
> > >
> > > On the one hand, we adapted the novel contributions of our work, placing greater emphasis on Dynamic Weighting and Label Sampling in the revised main paper.
> > >
> > > Also, we delved into the obstacles faced by the data-free adversarial knowledge transfer framework [6] in the heterogeneous FL setting. Specifically, while DFRD with [6] achieves excellent performance compared to baselines, we contend that scenarios characterized by high data heterogeneity continue to pose an obstacle to the performance improvement of DFRD with [6] (refer to Table 1). This is because it only dominates the contributions among different clients based on the sample proportion (i.e., $\tau_{i, y}$), which constrains the low-end clients (i.e., those with fewer datasets) who would otherwise make unique contributions to model training. As demonstrated by existing efforts (e.g., [a] [b] and [c]), the more rational control of contributions among different clients is a systematic and formidable challenge, one that we plan to address in future work. We have updated the aforementioned content in the revised paper.
> > >
> > > We aspire for our response to address the reviewer's concerns. We remain dedicated to addressing concerns you may possess with utmost eagerness.
> > >
> > > [a] Zhang, Jie, et al. "Federated learning with label distribution skew via logits calibration." International Conference on Machine Learning. PMLR, 2022.
> > >
> > > [b] Luo, Mi, et al. "No fear of heterogeneity: Classifier calibration for federated learning with non-iid data." Advances in Neural Information Processing Systems 34 (2021): 5972-5984.
> > >
> > > [c] Shen, Zebang, et al. "An agnostic approach to federated learning with class imbalance." International Conference on Learning Representations. 2021.

---

> > > > ### Author Response · Authors · 2023-08-16
> > > >
> > > > Dear reviewer. I kindly inquire whether our response has successfully addressed your uncertainties. We remain dedicated to addressing concerns you may possess with utmost eagerness.

---

### Official Review · Reviewer_Yxnp · 2023-07-08

**Soundness:** 3 good
**Presentation:** 3 good
**Contribution:** 3 good
**Rating:** 5
**Confidence:** 4

**Summary:**

This paper proposes a new method called DFRD for robust and privacy-constrained Federated Learning (FL) in the presence of data and model heterogeneity. DFRD uses a conditional generator to approximate the training space of local models and a data-free knowledge distillation technique to overcome catastrophic forgetting and accurately extract knowledge from local models. The experiments show that DFRD achieves performance gains compared to baselines.

**Strengths:**

1. The writing is easy to follow

2. The overall idea seems novel

3. The authors justify the necessities of loss functions and validate their effectiveness in the experiment.

**Weaknesses:**

1. This work is well-engineered, comprising multiple components and hyperparameters. Although stability testing was performed on the same dataset, the optimal hyperparameters for different datasets appear to be unstable (see Fig 5).

2. Although the authors propose loss functions aiming to improve fidelity and diversity, the synthetic images provided in the appendix are still far from achieving these goals. It is also questionable whether such characteristics are really necessary for downstream KD and ensembling.

3. The EMA strategy to avoid forgetting is not new in continuous learning, so the novelty of this technique is limited in this paper.

4. Based on the presented learning curves, the models do not seem to converge after 100 communication rounds for both the proposed and benchmark methods. This undermines the persuasiveness of the results reported in the tables.

5. The overall performance is very low, e.g., less than 20% accuracy for 10-class classification. I have concerns about the practicality of the method given its complexity and unsatisfactory performance in tackling the challenging non-iid setting.

6. I appreciate the authors' efforts to conduct numerous experiments, but how the settings overlap between experiments is not clear. It would be helpful if the authors could provide a table to show the connections. For example, I found it difficult to relate the values in the figures to the numbers in the tables. Ideally, the same experimental setting would result in the same number.

Minor comments:

The pseudocode provided in the appendix is not well organized. There seems to be a convoluted operation between server and client.


**Questions:**

1. As this method already uses EMA on global model (G) to get \tilde{G}, why does it need another step to find KL divergence on both G and \tilde{G}?

2 How the w, \rho, and \delta play roles in data heterogeneity?

3 There is a line of work on KD for FL aggregation. Could the author justify why " FedFTG [48] and DENSE [49], which are the most relevant methods to our work?" comparing the other related work mentioned in the manuscript (such as [27], [40], [47])  and Wu et al. listed below:

Wu, C., Wu, F., Lyu, L., Huang, Y., & Xie, X. (2022). Communication-efficient federated learning via knowledge distillation.


**Limitations:**

Limitations are discussed in appendix

---

> ### Author Rebuttal · Authors · 2023-08-08
>
> We highly appreciate Reviewer Yxnp for positive comments and precious feedback on our work. We respond to specific comments below.
>
> Q1: This work is well-engineered, comprising multiple components and hyperparameters. Although stability testing was performed on the same dataset, the optimal hyperparameters for different datasets appear to be unstable (see Fig 5).
>
> R1: Thanks for pointing out the details of Fig 5. We argue that there is no correlation between optimal hyperparameters for different datasets. Intuitively, different datasets correspond to different learning tasks, so it is difficult for any method (including DFRD) to ensure consistent optimal hyperparameters for varying learning tasks during practical experiments.
>
> Q2: Although the authors propose loss functions aiming to improve fidelity and diversity, the synthetic images provided in the appendix are still far from achieving these goals. It is also questionable whether such characteristics are really necessary for downstream KD and ensembling.
>
> R2: These images are necessary for downstream KD and ensembling, evidenced by Tables 1 and 2, that is, these images significantly augment the global model. Also, we present synthetic images under different diversity constraints (mul, add, cat, n-cat, none) in the Appendix (see Fig.14 - 18), where mul is proposed by us (see Fig. 14). The synthetic images exhibited in Fig. 14 clearly outperform other synthetic images (see Figs. 15-18) in terms of fidelity and diversity.
>
> Q3: The EMA strategy to avoid forgetting is not new in continuous learning, so the novelty of this technique is limited in this paper.
>
> R3: We agree that the EMA strategy is not new in continuous learning, but our innovation is that we are the first effort to apply the EMA strategy to heterogeneous FL with data-free knowledge distillation.
>
> Q4: The overall performance is very low, e.g., less than 20% accuracy for 10-class classification. I have concerns about the practicality of the method given its complexity and unsatisfactory performance in tackling the challenging non-iid setting.
>
> R4: In our work, we set up extensive heterogeneity settings, including data heterogeneity and model heterogeneity. In terms of data heterogeneity (i.e., non-iid settings), the accuracies of FedAvg and FedAvg+DFRD (+DENSE and +FedFTG) are much greater than 20% for different non-iid settings (including the challenging non-iid setting, i.e., $\omega=0.01$) under model homogeneity, see Table 1. In Table 2, we specifically investigate challenging scenarios characterized by both data and model heterogeneity, resulting in a considerable number of accuracies falling below the 20% threshold, especially on Tiny-ImageNet and FOOD101. Notably, the accuracy of DFRD is greater than 20% on SVHN and CIFAR-10 in most cases. Therefore, we are confident in the practicality of our method.
>
> Q5: I appreciate the authors' efforts to conduct numerous experiments, but how the settings overlap between experiments is not clear. It would be helpful if the authors could provide a table to show the connections. For example, I found it difficult to relate the values in the figures to the numbers in the tables. Ideally, the same experimental setting would result in the same number.
>
> R5: Due to the space limitation of the main paper, we detail experimental settings in Appendix E. We also agree with the reviewer's comments and will update them in the revised paper.
>
> Q6: The pseudocode provided in the appendix is not well organized. There seems to be a convoluted operation between server and client.
>
> R6:  There are no convoluted operations between server and clients. For ease of understanding, we will add a description of the pseudocode in the revised paper.
>
> Q7: As this method already uses EMA on global model ($G$) to get $\tilde{G}$, why does it need another step to find KL divergence on both $G$ and $\tilde{G}$?
>
> R7: In our paper, $\tilde{G}$ stores previous knowledge learned from the local models and is used as a complement to $G$ in DFRD. $\tilde{G}$ does not contain information about $G$ in the latest communication round. See lines 19 to 22 in Algorithm 1 for detail. Therefore, an additional step is required to compute the KL divergence for both $G$ and $\tilde{G}$.
>
> Q8: How the $\omega$, $\rho$, and $\sigma$ play roles in data heterogeneity?
>
> R8: In our paper, Dirichlet process $Dir(\omega)$ is utilized to assign training data to each client, which is frequently used in existing federated learning works [34,37, 38]. Here, $\omega$ is the concentration parameter of $Dir(\omega)$ and smaller $\omega$ corresponds to stronger data heterogeneity. We provide the visualizations of the data partitions for the six datasets at varying $\omega$ values, which can be found in Fig.6 and 7 in Appendix. In addition, $\rho$ and $\sigma$ are parameters related to model heterogeneity rather than data heterogeneity, see Appendix D for detail.
>
> Q9: There is a line of work on KD for FL aggregation. Could the author justify why " FedFTG [48] and DENSE [49], which are the most relevant methods to our work?" comparing the other related work mentioned in the manuscript (such as [27], [40], [47]) and Wu et al. listed below: Wu, C., Wu, F., Lyu, L., Huang, Y., & Xie, X. (2022). Communication-efficient federated learning via knowledge distillation.
>
> R9: To ensure fair comparisons, we neglect the comparison with methods that require to download auxiliary models or datasets, such as RHFL[27], FedDF[40] and FedGen [47] and the mentioned paper.  Also, we'll add the mentioned paper as one of our references.

---

> > ### Author Response · Authors · 2023-08-16
> >
> > Dear reviewer. I kindly inquire whether our response has successfully addressed your uncertainties. We remain dedicated to addressing concerns you may possess with utmost eagerness.

---

### Author Rebuttal · Authors · 2023-08-08

We thank all reviewers for their thoughtful, constructive, and positive review of our manuscript. We are encouraged to hear that the reviewers found the DFRD method we present to be interesting and practical (Reviewers i49s, 1PRp), and thoroughly-evaluated (Reviewers Yxnp, 1PRp, XBHz, i49s, deo9). Meanwhile, they view our methodology as novel (Reviewers Yxnp, 1PRp) and our manuscript as well-written (Reviewers Yxnp, 1PRp). In response to the feedback, we provide detailed responses to address each reviewer's concerns below.

---

### Decision · Program_Chairs · 2023-09-21

**Decision:**

Accept (poster)

**Comment:**

Reviewers found this paper exciting and unanimously recommended accept.